https://doi.org/10.1038/s42003-020-0938-9　　**OPEN**

# SwiftReg cluster registration automatically reduces flow cytometry data variability including batch effects

Jonathan A. Rebhahn[1], Sally A. Quataert[1], Gaurav Sharma[2,3] & Tim R. Mosmann[1✉]

Biological differences of interest in large, high-dimensional flow cytometry datasets are often obscured by undesired variations caused by differences in cytometers, reagents, or operators. Each variation type requires a different correction strategy, and their unknown contributions to overall variability hinder automated correction. We now describe swiftReg, an automated method that reduces undesired sources of variability between samples and particularly between batches. A high-resolution cluster map representing the multidimensional data is generated using the SWIFT algorithm, and shifts in cluster positions between samples are measured. Subpopulations are aligned between samples by displacing cell parameter values according to registration vectors derived from independent or locally-averaged cluster shifts. Batch variation is addressed by registering batch control or consensus samples, and applying the resulting shifts to individual samples. swiftReg selectively reduces batch variation, enhancing detection of biological differences. swiftReg outputs registered datasets as standard .FCS files to facilitate further analysis by other tools.

[1] David H. Smith Center for Vaccine Biology and Immunology, University of Rochester Medical Center, Rochester, NY, USA. [2] Department of Biostatistics and Computational Biology, University of Rochester, Rochester, NY, USA. [3] Department of Electrical and Computer Engineering, University of Rochester, Rochester, NY, USA. ✉email: tim_mosmann@urmc.rochester.edu

Several types of variability contribute to the changes in marker intensity (using fluorescence or mass labels) that are inherent in flow cytometry. Day-to-day (batch) variations in global channel values can be caused by cytometer settings (e.g., photomultiplier tube (PMT) voltages, laser power, or different cytometers). The fluorescence of positively stained cells can be affected by staining protocol variations, antibody batches, or reagent instability. More complex changes in multiple channels are induced by variables that affect cell health and viability, e.g., shipping, cell handling, thawing, processing, and operator variability. Fluorescence intensities may also be affected by biological variations, including genetics, environment, disease, age, gender, lifestyle, therapy, or microbiome. In any study, only one or a few of these sources of variation will be the target of investigation of the study—the others should be minimized so that the target variation (e.g., therapy-induced changes) can be analyzed clearly. Thus methods to reduce variability should ideally be objective, yet selective, for certain types of variability.

Manual gating analysis can be selectively adjusted to deal with some batch or individual variability, but the process is time-consuming and subjective. Some manual gating software packages offer auto-positioning gates to adjust for batch effects. However, these approaches are limited to two-dimensional (2D) gates, require a priori identification of gates, and fail in the presence of large shifts in subpopulation location.

There are now many excellent automated methods to identify subpopulations in flow cytometry samples. The strategies used to address inter-sample variation in such methods broadly fall into three categories: (1) Template-based methods generate models from selected or pooled samples and store all the relevant parameters (e.g., centroids, shapes, proportions, etc.)[1–6]. Individual samples can then be assigned to the template model. These methods work well if batch variation is smaller than the biological variation of interest, for example, our SWIFT algorithm[3] was able to handle substantial variations between analysis centers[7]. However, as batch variation increases, cells are assigned more frequently to the wrong clusters. (2) Cluster matching approaches attempt to match clusters between individually clustered samples to achieve a 1-to-1 or 1-to-$N$ mapping of clusters with similar characteristics across samples[8–16]. Two of these methods[15,16] try to mitigate adverse effects of batch variation during the cluster matching process using a random effects model during the matching process. (3) Registration (data alignment) approaches are normally preprocessing steps to move the data directly (i.e., register) to improve alignment across samples[17,18]. Of these three classes of approach, registration has the advantage that it is a preprocessing step that leaves open the possibility of subsequently analyzing the cell subpopulations by any of the wealth of flow analysis programs now available.

The registration (fluorescence normalization) programs fdaNorm and gaussNorm[17] normalize one channel at a time but require pre-gating of a subpopulation and do not address multidimensional linkages between biological subpopulations. A per-population "local" approach[18] builds upon fdaNorm, tightly integrating local (subpopulation specific) intensity normalization with the gating process. Specific features (histogram peaks or valleys) of manually gated or semi-manually selected data are used to modify samples to match a reference sample. However, this approach still relies on manual gating and does not provide an exhaustive registration of subpopulations at high resolution.

To address these issues, we have developed an automated, flexible registration tool, swiftReg, that uses the high-resolution cluster information from the SWIFT clustering algorithm as the basis for registration and generates free-standing registered data files that can subsequently be analyzed by any automated or manual analysis method. This approach has several advantages—first, cells are assigned to clusters using information from all channels, so even large shifts in one channel can be correctly identified and corrected because of the information in other channels. Second, the method should be robust to large changes in specific subpopulations, e.g., loss of CD4+ populations in AIDS, situations in which methods based on bulk channel shifts would result in the wrong adjustment. Third, this high-resolution registration tool can accommodate shifts of different magnitude or direction in many different subpopulations. Fourth, the swiftReg tool can be used in either channel-specific or fine-grained subpopulation-specific modes. As a result, swiftReg can selectively minimize batch variations, while preserving biological variations and thus allowing meaningful sample comparison with greater clarity.

## Results

**Identification of variation using SWIFT clusters.** As described above, several sources of variation may exist in flow cytometry data. The high-resolution SWIFT cluster templates provide sensitive tools for both identifying and then correcting different sources of variation. An initial SWIFT cluster template is produced from a reference sample, then any number of test samples are assigned to that template. Each cell is assigned to the most probable cluster, so often each template cluster will "catch" the appropriate cells even if that subpopulation has shifted substantially. However, the centroid of the resulting subpopulation may be shifted relative to the template. Figure 1a shows the variation in a fluorescence cytometry dataset (JMW090 and JMW092) of influenza peptide-stimulated human peripheral blood mononuclear cell (PBMC) samples[2,19]. The heatmaps show the correlations of cluster centroids between samples (see "Methods") in an experiment that contained variations due to assay day, cytometer, subject, and bleed. Major changes were caused by cytometer and assay day differences, whereas sequential blood samples from the same subject were much more consistent.

Quality control (QC) plots (Fig. 1b, c) show, in bleed 1 of the same samples, the variation of the centroids of each cluster (dots), each sample (columns), and each channel (rows). The horizontal line for each channel represents the 1:1 log ratio of the sample cluster median fluorescence intensity (MFI) to a standardized cluster MFI (standard values were the average of the MFIs for that cluster in all eight samples from subject 5). Thus a dot that lies further off the line indicates a greater MFI inconsistency between sample and standard for that cluster, and the pattern of dots for each sample is unique but may be similar to that of other samples that experienced similar experimental conditions. The degree of similarity is analogous to correlation measures, but dot patterns reveal more underlying complexity. For example, dot patterns for CD4 are more similar for samples run on the same cytometer, while patterns for CD45RA suggest subjects 1–4 are more similar to each other than to subjects 5 and 6.

The QC plots can facilitate rapid qualitative assessment of variation between sample groups when visualizing all channels and samples together (Fig. 1c). This fine-grained analysis shows that certain channels, e.g., CD45RA and CD4, contribute more strongly to MFI inconsistencies between samples than, for example, any of the scatter channels that all have comparatively tighter distributions about the 1:1 line. In addition to the variations in fluorescence intensities, there were substantial variations in the size (number of cells) of the clusters, potentially due to both mis-assignment of cells to the wrong cluster because of fluorescence changes, as well as genuine changes in subpopulation sizes.

Thus the SWIFT high-resolution cluster maps provide a very sensitive tool for identifying and localizing inter-sample variation.

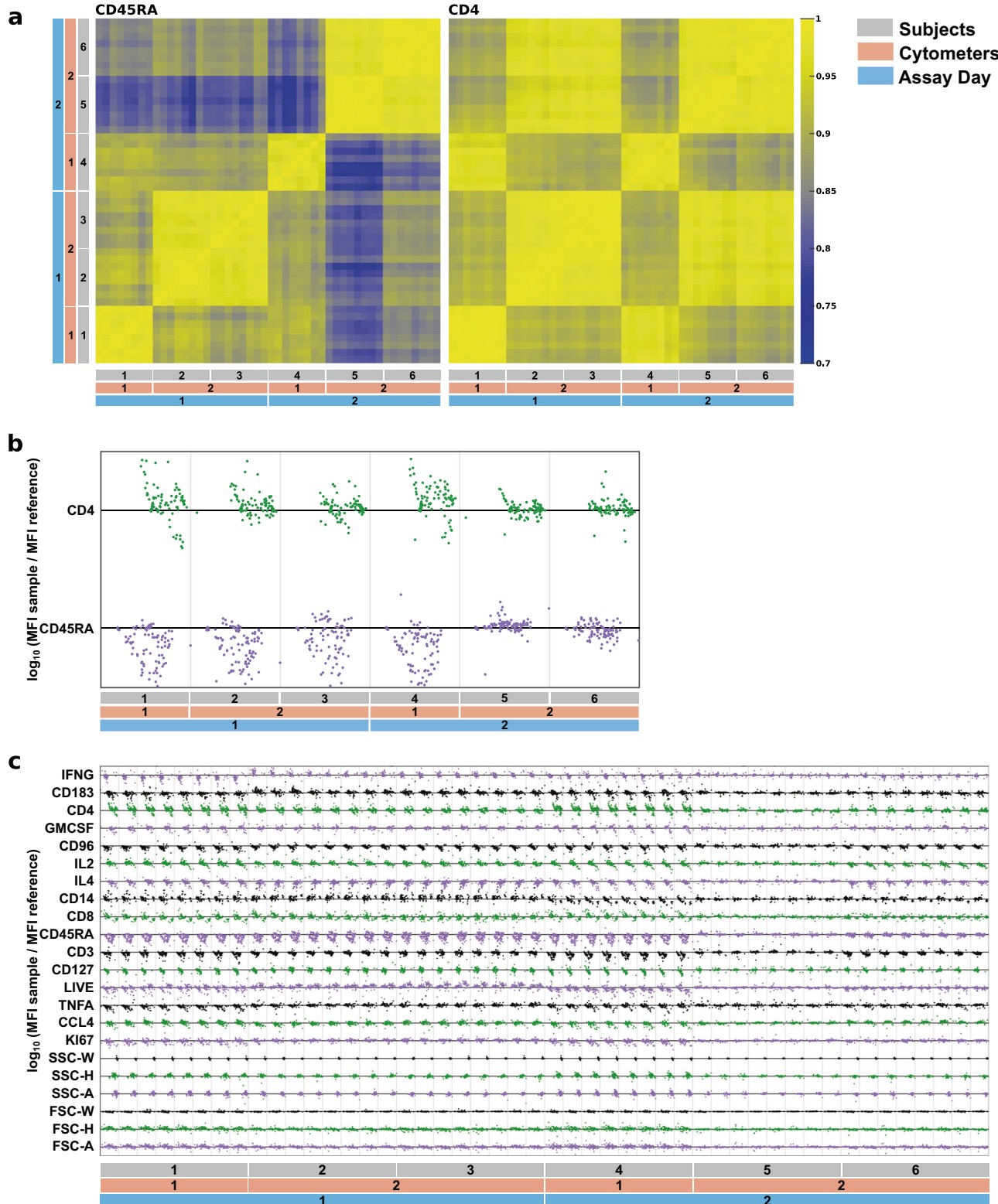

If this variation results in mis-assignment of cells to inappropriate clusters, then cluster registration should improve the identification of subpopulation size variation between the experimental groups.

**Strategy for SWIFT-based registration**. Some previous registration methods have registered fluorescence intensities between samples using all, or a major subset of the data, in single-channel histograms. However, a shift in the global fluorescence values in one channel could be due to a change in the proportions of bright and dim cell subpopulations (Fig. 2a), or a shift in fluorescence intensity. Different registration adjustments are required in these two situations. Therefore, we use the model defined in the SWIFT cluster template (i.e., cluster centroids, shapes, and proportions)

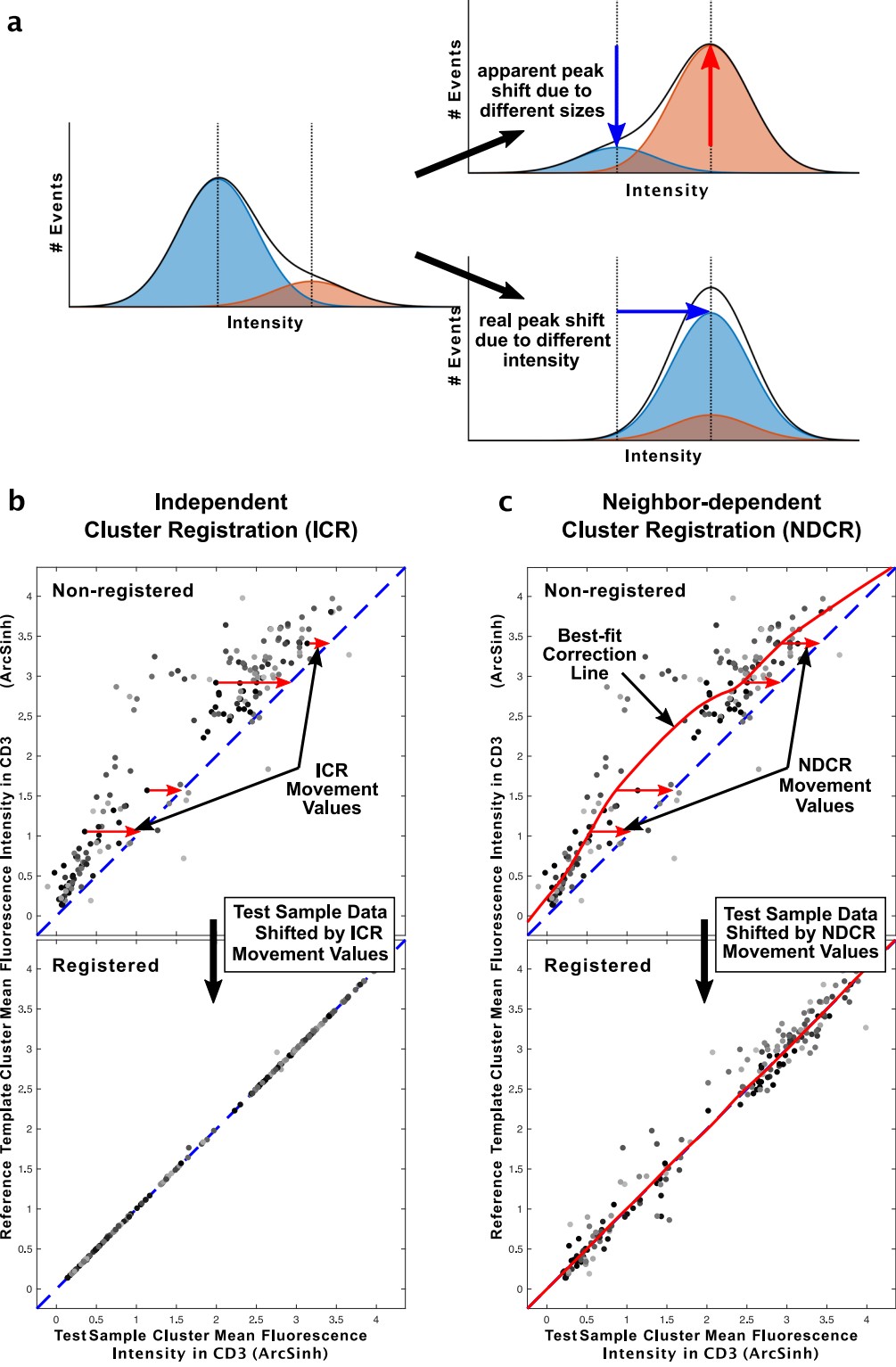

**Fig. 2 Strategy for SWIFT cluster-based registration. a** Two Gaussian distributions demonstrate two possible explanations for bulk population shifts. Compared to the middle panel, the peak shift in the overall population (black line) could be due to changes in the sizes or intensities of the constituent red and blue subpopulations. **b** Independent cluster registration (ICR): from JMW092, a test sample was assigned to a template containing 184 clusters derived from the reference sample and the cluster mean fluorescence intensities of test and reference samples compared (non-registered). For each cluster, the cluster movement value was calculated (arrows shown for a few selected clusters), and these values were applied to register the sample events and improve the correlation between sample and reference clusters (registered). The axes represent the ArcSinh-transformed mean fluorescence intensity values for each cluster, shown in the CD3 channel. **c** Neighbor-dependent cluster registration (NDCR): the cluster positions were determined as in **b**, and a best-fit correction line was calculated (red line, non-registered). All individual events were registered according to cluster movement values derived from this line, resulting in locally constrained registration that preserves local variation (registered). The axes are the same as in **b**.

that incorporates information from all other channels, to register samples at the subpopulation level.

From JMW092, a test sample was assigned to a reference template produced by SWIFT clustering of a reference sample. The cluster movement vectors between test sample and reference template were measured for each cluster, e.g., the staining in almost all clusters of the test sample was slightly lower than the reference sample, in the CD3 channel shown in Fig. 2. In independent cluster registration (ICR), each cell is moved (in several iterations) according to the cluster movement vector of its cluster. For cells with partial membership in more than one cluster, the cell is moved according to the weighted vector of the relevant clusters (Fig. 2b). As the cells are re-assigned to the reference template model after each iterative shift, a cell may change its majority membership to a different cluster during the registration process. Thus the resulting medians do not always reach the exact reference values after four iterations.

However, batch effects may cause mainly global changes in particular channels. For example, alterations in staining will alter the MFI of positively stained cells but not autofluorescence, whereas alterations in PMT amplification will alter both. Because of the unknown contributions of different sources of variation, we developed an empirical neighbor-dependent cluster registration (NDCR) method to register cells using a correction value determined by all clusters with similar values in that channel. A best-fit correction line for each channel is generated from the cluster information and each cell is then registered according to the cluster's position along this correction line (horizontal arrows, Fig. 2c).

**Evaluation of ICR and NDCR with semi-synthetic and real data.** To evaluate swiftReg's abilities to improve cell classification, we constructed semi-synthetic datasets to contrast two types of variation: subpopulation-specific shifts and bulk channel shifts. Cluster templates were derived from three samples in dataset JMW092, and one well-resolved, moderate-sized cluster of cells was selected from each sample. A semi-synthetic series of samples was then constructed for each cluster by moving the cluster progressively in two dimensions. A second series of samples was constructed, for each cluster, by moving all cells progressively in the same two dimensions. This strategy allowed unambiguous tracking of cells and clusters.

Samples in each semi-synthetic series were then assigned to the cluster template produced from the normal sample, and the cluster assignments of the target clusters were determined. As the magnitude of synthetic adjustments increased, cells in the original cluster were increasingly mis-assigned to the wrong cluster (illustrated for one cluster in Fig. 3a, which is Cluster One in Fig. 3b). Samples in each series were then registered (using ICR, NDCR, or both) to the first sample in the series. Registration substantially improved the ability to assign the target cells to the appropriate cluster (Fig. 3b). As expected, ICR was more effective at improving assignment when cells in only one cluster were moved, whereas NDCR was often more effective at correcting samples in which all cells were moved, presumably because information from many clusters provided a better estimate of the global shift. Importantly, sequential registration by NDCR then ICR provided the best, or close to the best, performance for each situation (Fig. 3b). Registration allowed cells to be assigned more robustly to the correct cluster, tolerating deviations that were up to sixfold greater (Fig. 3b).

Using real, unmodified flow data, we next evaluated the effect of different numbers of registration iterations, and two approaches to registration iterations, partial or full. The partial method performed an incomplete position-update per cell per iteration, e.g., 25%, 33%, 50%, and then 100% of the cluster movement vectors. The partial iterations were designed to test the possibility that correct assignment might be improved by a more cautious approach to the final positions. The full method performed a complete position-update per cell per iteration, thus for four iterations each step would be 100% of the cluster movement vectors.

From the SDY420 study in the publicly available ImmPort database (https://www.immport.org/shared/study/SDY420)[21], we chose five samples from the same subject (internal standards) that were analyzed by CyTOF on different days. One sample was randomly selected as the reference and all other samples were registered to it. Registration was tested using partial vs. full position-updates for NDCR, ICR, and both (NDCR followed by ICR). The number of iterations was also varied from 1 to 10 for NDCR or ICR, and to keep the total number of steps for the dual method equivalent, we used 1 to 5 for NDCR followed by 1 to 5 for ICR. Euclidean distances between registered clusters and reference clusters were measured, as well as differences between cluster sizes. The Root Mean Squared Errors (RMSE) of the distances and the sizes were calculated for registered samples, and then divided by the RMSE of the non-registered samples to yield a relative RMSE. For NDCR RMSEs, there is little difference between partial or full iterations, and errors do not improve substantially after the third iteration (Fig. 3c). For ICR, full iterations resulted in lower RMSEs than partial iterations and neither improves substantially after the fourth iteration. Registration by NDCR+ICR with full iterations resulted in slightly lower relative RMSE than partial iterations and neither improves substantially after two iterations of each (i.e., two NDCR followed by two ICR). Furthermore, the combined NDCR+ICR method gives the lowest relative RMSE for cluster sizes of all scenarios tested.

**Improvement of sample uniformity.** NDCR and NDCR+ICR were applied separately to the fluorescence flow cytometry data shown in Fig. 1, and the registered samples were assigned to the original cluster template. As expected, the cluster centroids in each channel correlated more closely between samples after registration, and consistent with Fig. 3, sequential application of NDCR+ICR achieved the best results (Fig. 4). Very importantly, the registration of the positions of each cluster also improved the correlation of cluster sizes between samples (Fig. 4, size), even though the registration did not directly target the cluster sizes. This is consistent with the ability of swiftReg to correct the mis-assignment of cells demonstrated in semi-synthetic data (Fig. 3b) and also with the reduction of the RMSE of the cluster sizes in CyTOF data (Fig. 3c).

**Comparison between swiftReg and a prior method.** The semi-automated method described by Finak et al.[18] uses operator input to define target subpopulations and then automatically registers the data in multiple samples to a reference. We compared automated swiftReg registration using the same dataset—a subset of a 10-color intracellular cytokine staining dataset in which pre- and post-vaccination PBMC samples from 48 subjects in a Phase I HIV vaccine trial (HVTN080 https://flowrepository.org/id/FR-FCM-ZZ7U) were stimulated with three HIV antigens[20]. Some batches in this dataset lacked positively stained, rare subpopulations for stimulated samples, thus certain subpopulations existed in some batches but not others. ICR should not be used in such cases so as to avoid potential mis-assignment. Therefore, NDCR alone was used to register all samples, and then activated T cell subsets were evaluated in the resulting data using the same gating strategy as

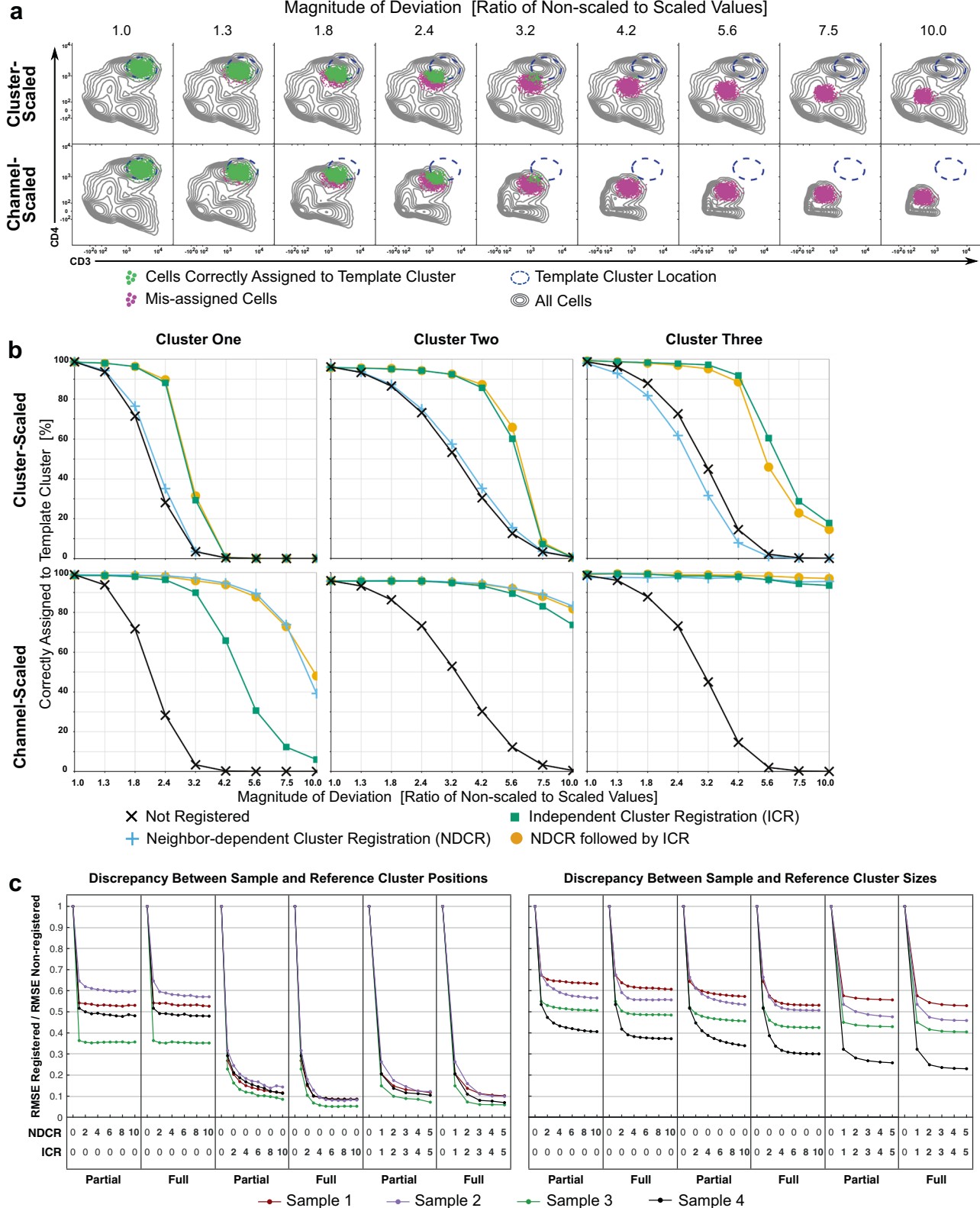

Finak et al. In this dataset, an apparently CD4+ Perforin+ cell subpopulation was identified as an artifact of day-to-day variation[18]. Cells in this subpopulation were enumerated in non-registered and registered samples by conventional bivariate gating (see Supplementary Fig. 1). The perforin staining varied between batches 880 and 1053 (Fig. 5a, reference and non-registered), but after registration the values in batch 1053

(registered) were aligned well with batch 880. Even though registration was performed without specifically targeting any subpopulation, swiftReg was able to markedly reduce the number of the CD4+ Perforin+ cells, without materially affecting CD4 T cell subpopulations producing Interleukin 2 and Interferon-gamma (Fig. 5b). Thus swiftReg improved the consistency of the dataset without operator guidance.

**Fig. 3 Evaluation of iterative registration using semi-synthetic and real data. a** One of the samples described in Fig. 1 was clustered using the SWIFT algorithm. To create semi-synthetic samples with cluster-scale variability, after compensation and background subtraction, values in two channels for one cluster were divided by scaling factors ranging from 1 (no change) to 10 in 8 steps. For semi-synthetic samples with channel-scale variability, all cells were scaled by the same eight increments. The increments are expressed as magnitudes, i.e., numbers increase to indicate larger deviations between non-scaled and scaled values. The data were then ArcSinh transformed (see axes). All samples were then assigned to the original template. Cells in the specified cluster were assigned correctly (green) or incorrectly (magenta). Gray contours represent all cells. **b** Three cluster- and channel-scaled samples were registered to the corresponding original, non-scaled samples by ICR, NDCR, or NDCR+ICR. Registered samples were then assigned to the original cluster templates. For three clusters (one from each sample, Cluster One is shown in Fig. 3a), the capture of cells by the correct cluster was measured for increasing magnitudes of deviation. **c** From SDY420, five samples from the same subject were collected and frozen, then analyzed by CyTOF on different days. One sample was randomly selected as the reference and clustered using the SWIFT algorithm, and then the remaining samples were registered to the resulting template. Registration was repeated with varying numbers of iterations of NDCR or ICR or both and either partial or full position updates per iteration. All registered samples were then assigned to the reference SWIFT cluster template. The RMSE of each registered sample relative to the RMSE of its non-registered counterpart are shown.

**Batch registration reduces batch but not subject variation**. The most troublesome source of variation in many large flow datasets is batch variation, due to changes in cytometer settings, reagents, or sample handling between batches analyzed on different days. We have extended swiftReg to target the elimination of these batch effects, while maintaining biological variation to the maximum extent possible.

A cluster template is produced from an internal standard sample in the reference batch, and the internal standard from every other batch is registered to this template. The cluster movement vectors for each cluster are saved in a batch registration template and applied to each individual sample in the reference batch (the principle of batch registration is illustrated with synthetic data in Fig. 6a–f). If internal standards are not available, batch consensus samples can be used instead, provided that each batch includes a similar representation of all biological groups. This process reduces the overall differences between the two batches but does not affect variation between samples within a batch.

Batch registration reduces the variation between experimental groups, and this often increases the ability to detect differences between biological groups. In the publicly available ImmPort database (https://www.immport.org/shared/study/SDY420), the SDY420 study[21] describes the CyTOF flow cytometry data from 260 subjects with ages varying between 41 and 90 years, analyzed with a panel of 27 markers focusing on surface antigens of lymphocytes and myeloid cells. These samples were analyzed in 24 batches, with an experimental design that included an internal standard in each batch. Using SWIFT clustering and QC plots (described in Fig. 1) to examine the consistency of each parameter across samples, there were clearly batch effects between the internal standard samples (Fig. 7a, non-registered) and between subject samples. Batch registration was performed using the internal standards from each batch. This resulted in almost complete alignment of the internal standard samples (Fig. 7a, registered) and marked improvements in the consistency of marker expression in the young/old study samples (Fig. 7b, batch-registered). Note that heterogeneity within a batch was preserved, e.g., sample-specific variability in CD20 in one sample within the batch (blue arrows) or CD45RA in another sample (red arrows). The alignment of internal standards is further shown for five selected batches as stacked histograms before and after registration (Supplementary Figs. 2 and 3). For the same five batches, the remaining samples are also shown before and after batch registration (Supplementary Figs. 4 and 5).

A consensus registered sample was then created by subsampling and concatenating all registered samples, and a cluster template was created by clustering in SWIFT. All registered and non-registered samples were assigned to this template. In the non-registered data, 31 clusters showed differences between young and old subjects

(Fig. 8a, non-registered) even after Benjamini–Hochberg correction for multiple outcomes[22]. In contrast, 127 clusters showed young/old differences in the registered data (Fig. 8a, batch-registered). Thus registration (using only batch information) increased the ability to detect biological differences between the experimental groups.

To facilitate comparisons with previous analyses, we aggregated clusters by cluster gating (i.e., gating all cells within a cluster only according to the median fluorescence values of their cluster, Supplementary Fig. 6a) as described previously[3] into well-recognized phenotypes. As expected from the original analysis of the SDY420 dataset[21], B cell and CD8 T cell clusters were prominent among the clusters showing differences with aging (Fig. 8b, c). The SWIFT high-resolution analysis separates any subpopulations that would be multimodal in any combination of dimensions, resulting in large numbers of clusters. Figure 8d shows the detailed analysis of the cluster sizes in all subjects, for two selected naive CD8+ T cell clusters, as well as the total of all naive T cell clusters. The aggregated naive CD8 T cell subpopulation showed a clear downward trend with age, consistent with the analysis in the original study[21] in which the naive CD8 T cells showed the most prominent effect with age, and the variation in the data at different ages was similar in registered or non-registered data. As the naive CD8+ T cells are defined by several markers with strongly bimodal distributions, the probabilistic assignment process in SWIFT can overcome moderate batch effects and successfully assign cells to the correct broad phenotype, due to the multi-channel consideration of cluster membership. In contrast, more subtle variations, such as those defining cluster 322, show a stronger requirement for registration before clear-cut trends can be identified.

SWIFT separates clusters on the basis of multidimensional unimodality and therefore identifies subpopulations that cannot be defined in traditional 2D plots. Biological significance of such high-dimensional separations is supported by the behavior of clusters such as 124, which shows very little change with age in spite of being defined by markers similar to the age-dependent cluster 322 (Fig. 8d).

To further demonstrate the improvement of detection of biological differences in registered data using another analysis modality, the registered files were analyzed by manual gating using a single set of gates for all samples (Supplementary Fig. 6b). Both naive CD4 and naive CD8 T cell subpopulations showed more variation in the original data but less variation after registration (Fig. 9, age). This resulted in a much more obvious trend for reduced numbers of naive CD8 T cells with age and even revealed a marginally significant ($p$ value 0.0311) effect of age on the naive CD4 T cell subpopulation. These improvements in the detection of biological differences were accompanied by blunting, but not complete removal, of the batch effects demonstrated in the same data plotted according to batch number (Fig. 9, batch).

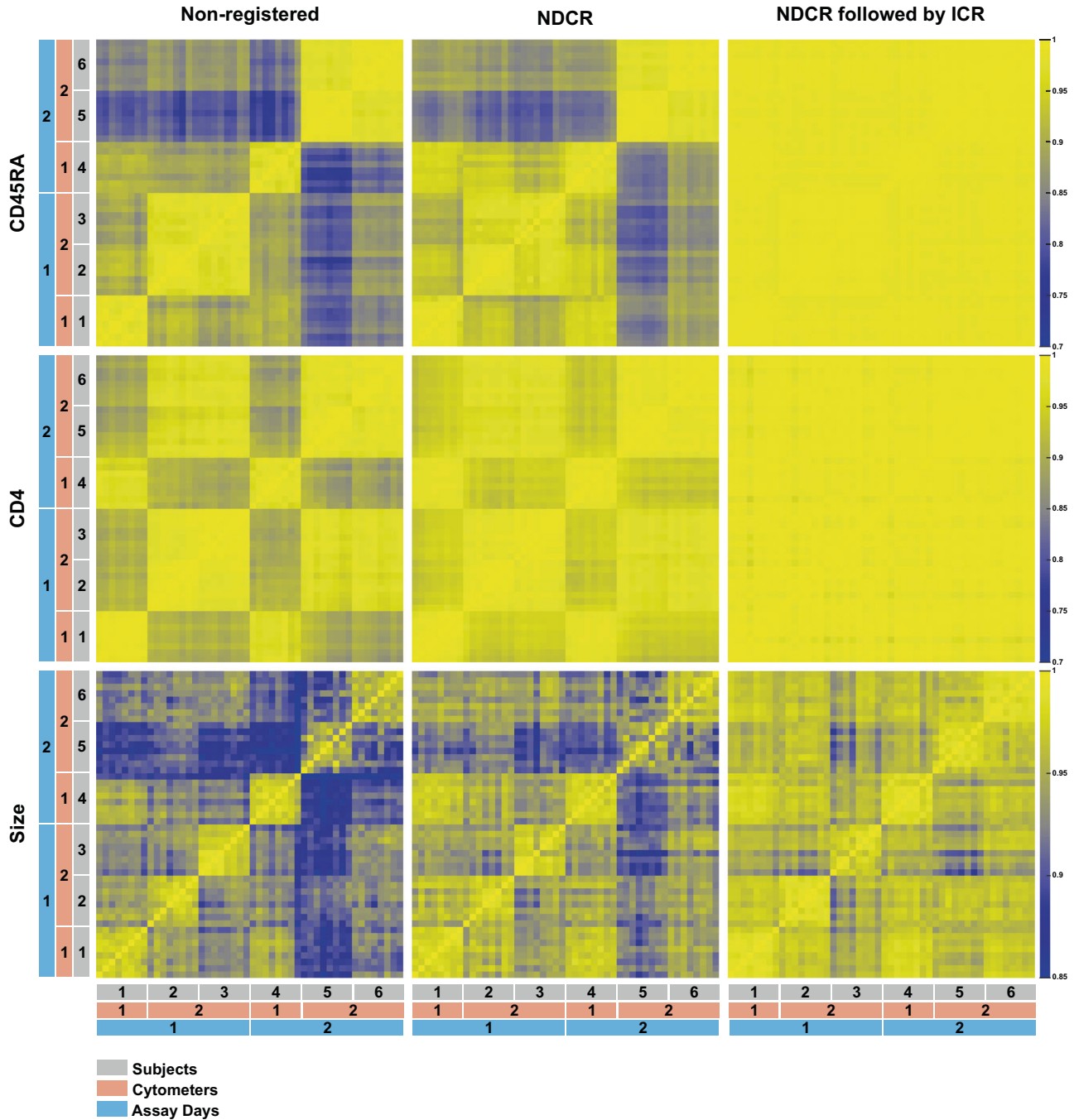

**Fig. 4 Registration improves sample uniformity.** The samples described in Fig. 1 were registered to the cluster template of Fig. 1. The heatmaps show Pearson correlations between all clusters in each sample pair for CD45RA, CD4, and the cluster sizes. Non-registered, NDCR, and NDCR+ICR results are shown.

## Discussion

swiftReg has several advantages. Registration using swiftReg is fully automated and does not require user selection of landmarks or target subpopulations. Several types of variation can be addressed, and batch registration can enhance identification of biological effects by selectively removing batch effects that would otherwise obscure small biological effects. A requirement for registration can be determined by examining the full dataset using tools such as the correlation heatmaps or QC plots described in Fig. 1. A particular advantage is that registration can be performed without any pre-definition of subpopulations of interest. In fact, it is critically important that the subpopulations of interest are not pre-defined, so that the registered dataset can be explored in a non-biased manner. We identified batch effects in all multi-batch datasets that we analyzed, from multiple laboratories, so swiftReg is likely to be helpful in many datasets.

When is swiftReg most required? In general, major cell types, defined by multiple clearcut markers (e.g., T cells are CD3+ CD4+CD20−CD14−CD19−) can be identified well by the basic SWIFT clustering assignment step, so that substantial batch variations can be handled effectively by the main SWIFT algorithm. Registration is more important for more subtle sub-population differences in which critical markers are not as distinctly bimodal, e.g., manually gated naive CD4 and CD8 T cell subpopulations, or some of the high-resolution SWIFT clusters.

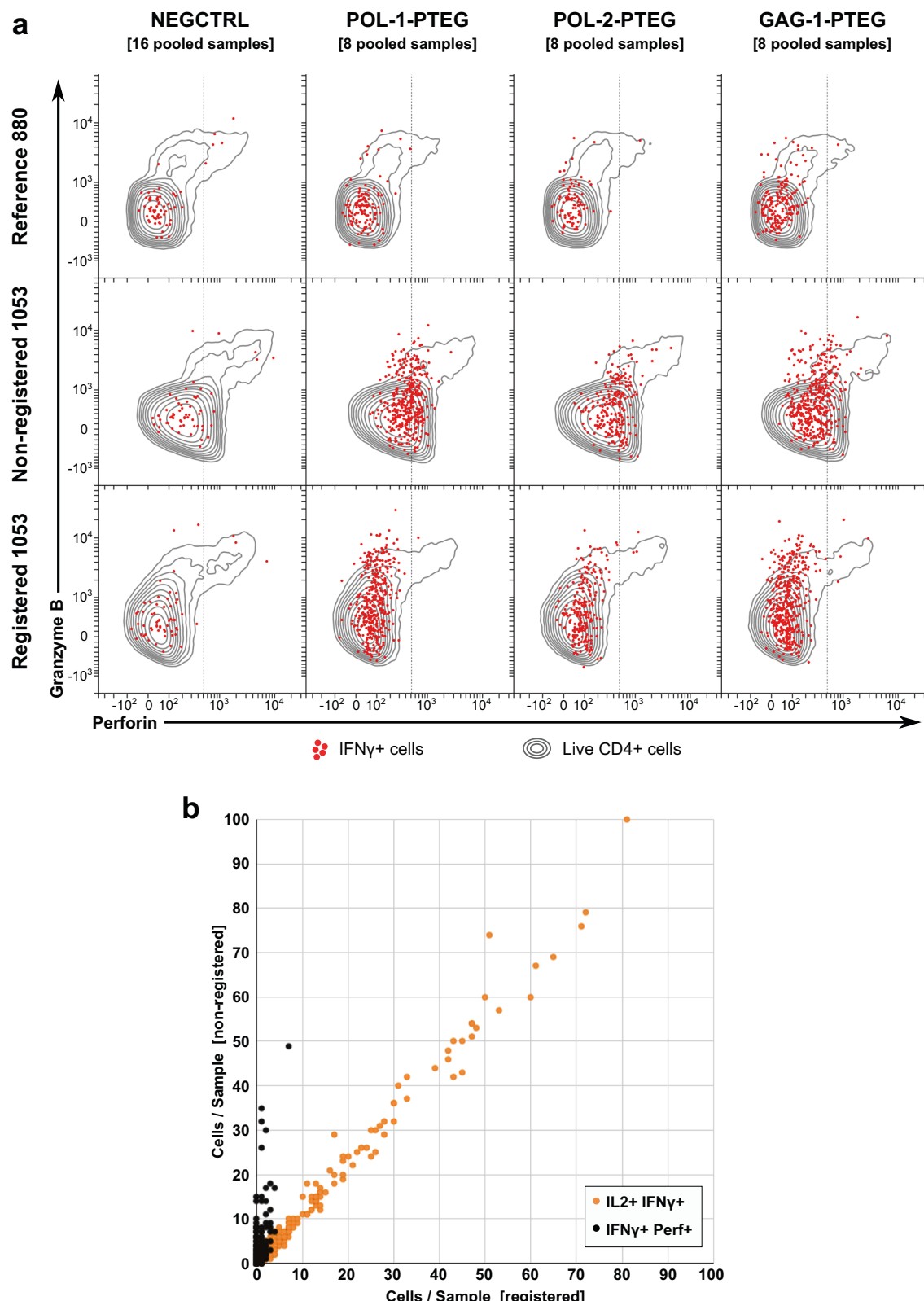

**Fig. 5 Registration reduced false positives of misaligned HVTN samples.** A SWIFT cluster template was produced from a concatenate of the antigen-stimulated samples from batch 880 in the fluorescence cytometry HVTN study HVTN080 (available in FlowRepository). All individual samples (total 435) were registered to this template by NDCR. Non-registered and registered samples were then gated identically (see Supplementary Fig. 1). **a** Data for each batch were aggregated (pooled for display) according to stimulation. Live, CD4+ cells are shown in gray contours and Interferon-gamma+ cells as red dots for non-registered reference batch 880, non-registered batch 1053, and registered batch 1053 samples. **b** CD4+ Interferon-gamma+ Interleukin-2+ (orange) and CD4+ Interferon gamma+ Perforin+ (black) cell counts are shown for all non-registered vs. registered samples.

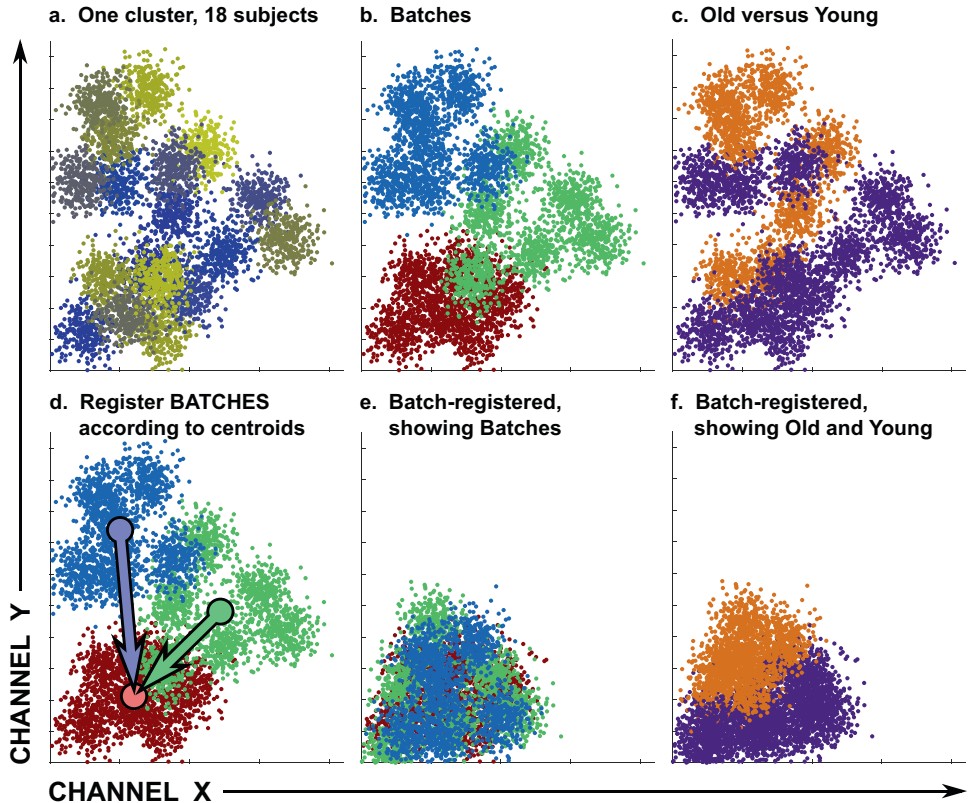

**Fig. 6 Strategy for selective removal of variation due to batches. a–f** The batch registration strategy is illustrated with synthetic data that models a single cluster as a two-dimensional Gaussian distribution. Eighteen Gaussians of 300 cells each represent 18 subjects, each with a different cluster centroid. Each subject is represented by a different color in **a**, and the three batches are designated in **b**. Six samples were designated as old and 12 as young (**c**). The cluster movement vectors necessary to register the centroids of the blue and green batches onto the red batch are shown in **d**, and the batches and young/old samples after registration are shown in **e**, **f**. Colors represent different subjects (**a**), batches (**b**, **d**, **e**), or age (**c**, **f**).

swiftReg can substantially improve data by reducing variability and particularly batch effects, but there are limits beyond which registration cannot help. Cluster registration is a supplement, not a replacement, for good experimental design and data reproducibility. If clusters are shifted substantially, there will inevitably be ambiguities in the assignment of cells to clusters, and neither manual nor automated registration can correct the aberrations. Although swiftReg can improve the overall detection of biological differences in datasets, we cannot exclude the possibility that occasional subpopulations could show anomalous results. If a particular subpopulation is absent in the reference sample, or absent from an entire batch, this subpopulation may not be aligned correctly by registration (see recommendations below).

The choice of reference sample for derivation of a cluster template is strongly dependent on the experimental design. This sample should include representatives of all subpopulations in the experiment (e.g., only antigen-stimulated samples contain activated T cell subpopulations) to ensure all subpopulations are registered appropriately. If unique subpopulations are suspected in some samples, then a concatenate of all samples should be used as reference. Batch registration should be performed with internal standards if available, so that the appropriate registration vectors for each cluster can be calculated without any influence from the biological groups in the experiment. Alternatively, batch concatenates should be used, provided biological groups are represented at similar proportions in each batch.

swiftReg is a flexible tool, with several choices for different types of experimental data. We envisage that batch registration will be the most widely used application of swiftReg. The use of NDCR and ICR will be influenced by the type of variation present in each dataset, but in general the most effective protocol is to use a few cycles of NDCR followed by a few cycles of ICR. The initial constrained NDCR cycles should correct global effects and leverage all the information from surrounding clusters to determine the initial correction vectors, followed by the fine tuning with ICR. In rare cases when a key subpopulation is missing from an entire batch, only NDCR should be used. Recommendations for the choice of method are:

- For batch registration with internal standards, use NDCR followed by ICR.
- To reduce all MFI variation due to either artifactual or biological differences between samples, e.g., enumerating cytokine-producing cells, use NDCR followed by ICR on individual samples.
- To remove channel-specific effects (e.g., staining or PMT voltage changes) in the absence of internal controls and to preserve minor positional information, use NDCR only.
- If some batches contain subpopulations that are completely absent in other batches (or their controls), use NDCR only.
- In each case, a total of four iterations (two each NDCR and ICR for the combined method) are recommended, as a compromise between speed and completeness.

The outputs from swiftReg are user-friendly, standard-format FCS 3.0 files in which each event has been shifted. These files can be analyzed either manually or by any of the automated clustering and gating methods that are becoming increasingly available. The production of standard .FCS files also makes swiftReg a useful component of automated processing pipelines. In all cases, the registered data from swiftReg should improve the quality of

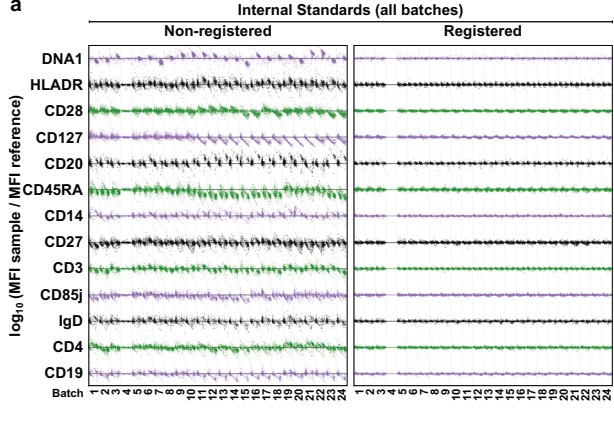

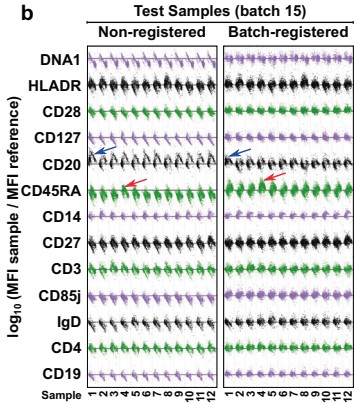

**Fig. 7 Evaluation of selective removal of variation due to batches.** Batch registration of the SDY420 study was performed by registering each of the internal standards to a reference (Batch 4) internal standard to generate 24 batch-specific registration templates, and the shifts in these templates were applied to the appropriate batches of individual samples (total 260 samples). For three batches, consensus reference samples were used because internal standards were unavailable. A concatenate of the registered samples was clustered in SWIFT, and all samples were assigned to the resulting cluster template. QC plots (described in Fig. 1) show the variation in the centroids of 13 selected channels of each cluster in all internal standards before and after registration (**a**) and in 12 actual samples from one batch before and after batch registration (**b**). Arrows indicate samples that are outliers for CD20 or CD45RA cluster centroids.

subsequent analysis and increase the detectability of biological changes.

## Methods

**Heatmap of cluster correlations**. The SWIFT clustering algorithm[2,3] written in MATLAB (The MathWorks, Inc., Natick, MA, United States) was used to produce a cluster template from a concatenate of influenza peptide-stimulated and negative control samples from a normal human subject. Forty-eight individual samples were assigned to this template as previously described[2]. This provided a table of the sizes (cells per cluster) of each sample and a second table of the centroids (i.e., medians of compensated fluorescence intensity values that are then ArcSinh-transformed) of each cluster, each channel, each sample. For cluster sizes or channel centroids, correlations were calculated between each sample pair $\mathcal{A}$ and $\mathcal{B}$, according to Eq. 1,

$$\rho(\mathbf{A}, \mathbf{B}) = \frac{1}{N-1} \sum_{i=1}^{N} \left( \frac{\mathbf{A}_i - \mu_{\mathbf{A}}}{\sigma_{\mathbf{A}}} \right) \left( \frac{\mathbf{B}_i - \mu_{\mathbf{B}}}{\sigma_{\mathbf{B}}} \right), \qquad (1)$$

where $N$ is the number of clusters, $\mathbf{A}$ and $\mathbf{B}$ are vectors of either single-channel cluster centroids or cluster sizes from sample $\mathcal{A}$ and $\mathcal{B}$ (respectively), $\mu$ and $\sigma$ are the mean and standard deviation of the respective vectors, and $\rho$ is the Pearson correlation coefficient (i.e., linear dependence) between vectors of samples $\mathcal{A}$ and $\mathcal{B}$[23].

**Creation of two sets of semi-synthetic samples**. A sample was clustered by SWIFT and one cluster was selected for tracking. The selected cluster was well resolved, meaning for repeated SWIFT assignments the cluster contained at least 80% of the original cells at least 80% of the time. We further selected the most stable cohort (>95% same cells, >95% of repeat assignments) of cells within the well-resolved cluster for precise tracking. The sample was then analyzed manually in FlowJo to identify background staining levels in each channel. The semi-synthetic sample was then compensated and the backgrounds subtracted. Then the CD3 and CD4 values of only the stable cohort of cells were divided by a series of scaling factors ranging from 1 (no change) to 10, in 8 steps, and the non-cohort cells were unchanged. Then the backgrounds were added back, and a new FCS file was generated for each scaling factor used. A second set of semi-synthetic samples was created by scaling CD3 and CD4 values of all cells. In both sets, the same stable cohort of cells was tracked.

**Selection of reference samples**. Registration broadly requires a reference (target) sample and test samples (to be registered). The reference is a single-sample or consensus (i.e., concatenation of multiple files) FCS 3.0 formatted file. The best reference sample is highly dependent on the dataset and should ideally contain all the subpopulations present in any sample to be registered, e.g., antigen-stimulated samples will contain additional subpopulations relative to negative controls. Consensus references reduce the risk of missing subpopulations within a dataset and minimize effects of occasional outliers. However, consensus samples will broaden peaks and decrease resolution, and inclusion of substantially different samples may result in false heterogeneity, i.e., one subpopulation represented by two peaks.

**Calculation of cluster movement vectors**. SWIFT was used to produce a cluster template from a reference sample. This template contains a Gaussian Mixture Model (GMM) that encodes each cluster in the reference sample as a set of centroids (mean of the transformed intensity of all events in the cluster in each dimension), covariances, and relative sizes. Test samples were assigned to the reference GMM, with every event allocated proportionally to one or more clusters, specifically, the proportion of an event allocated to a cluster equals the GMM posterior probability estimate that the event comes from the cluster. After initial assignment, new cluster centroids were calculated for the test sample. Individual channel movement values were calculated as the difference between the centroids of the sample and reference for that cluster. The cluster movement vector is the combination of the individual channel movement values, i.e., the cluster movement vectors will bring the centroids of each test sample cluster into alignment with those of the corresponding reference cluster.

**Independent cluster registration**. The overall goal of ICR is to align each test sample cluster, independently, to the corresponding cluster in the reference template. Thus each cell in the test sample is moved according to its cluster movement vector, to align the data more closely with the reference. If a cell has a very high probability of belonging to only one cluster, the movement vector is simply the cluster movement vector for that cluster. If a cell has partial membership in more than one cluster (e.g., 40% and 60% in A and B, respectively), the cell is moved along a compound vector assembled from the weighted vectors for each relevant cluster (40% A, 60% B). This avoids introducing discontinuities in the cell movements.

In samples requiring registration, some cells will almost certainly be assigned to the wrong cluster during the initial assignment to the reference template. To provide opportunities for correction of these initial mis-assignments, cells are moved iteratively, recalculating new centroids and cluster movement vectors after each iteration. The final cell intensities in each channel are then saved in a standard FCS file that can be analyzed by standard manual or algorithmic methods.

It is important to note that ICR can potentially introduce false heterogeneity if there are rare subpopulations that are only present in either the reference or the sample. This issue can be addressed using the following methods.

**Neighbor-dependent cluster registration**. To establish neighbor-dependent channel movement values, sample clusters are first ranked in ascending order according to the cluster centroid values in that channel, of the sample being registered. Because estimation uncertainty of centroids is inversely related to cluster size, clusters are weighted between 0 and 1 according to size (see below and Supplementary Fig. 7). For each channel, clusters are then binned such that the cluster weight per bin is a percentage of the total cluster weights in the sample. Thus, for example, with 20 bins, each bin would contain 5% by weight of clusters. The number of bins used is a function of the number of clusters (see below and Supplementary Fig. 8). The average bin intensity is the weighted average of the sample cluster centroids in that bin. The bin movement value is the weighted average of the differences between the sample and reference channel values for the clusters in that bin. Then a smooth spline curve is fitted across the average bin intensities (x-axis) and bin movement values (y-axis) and attenuated to zero for outlier regions with no clusters. For each channel, each cluster's neighbor-dependent movement value is then given by the curve at the point that

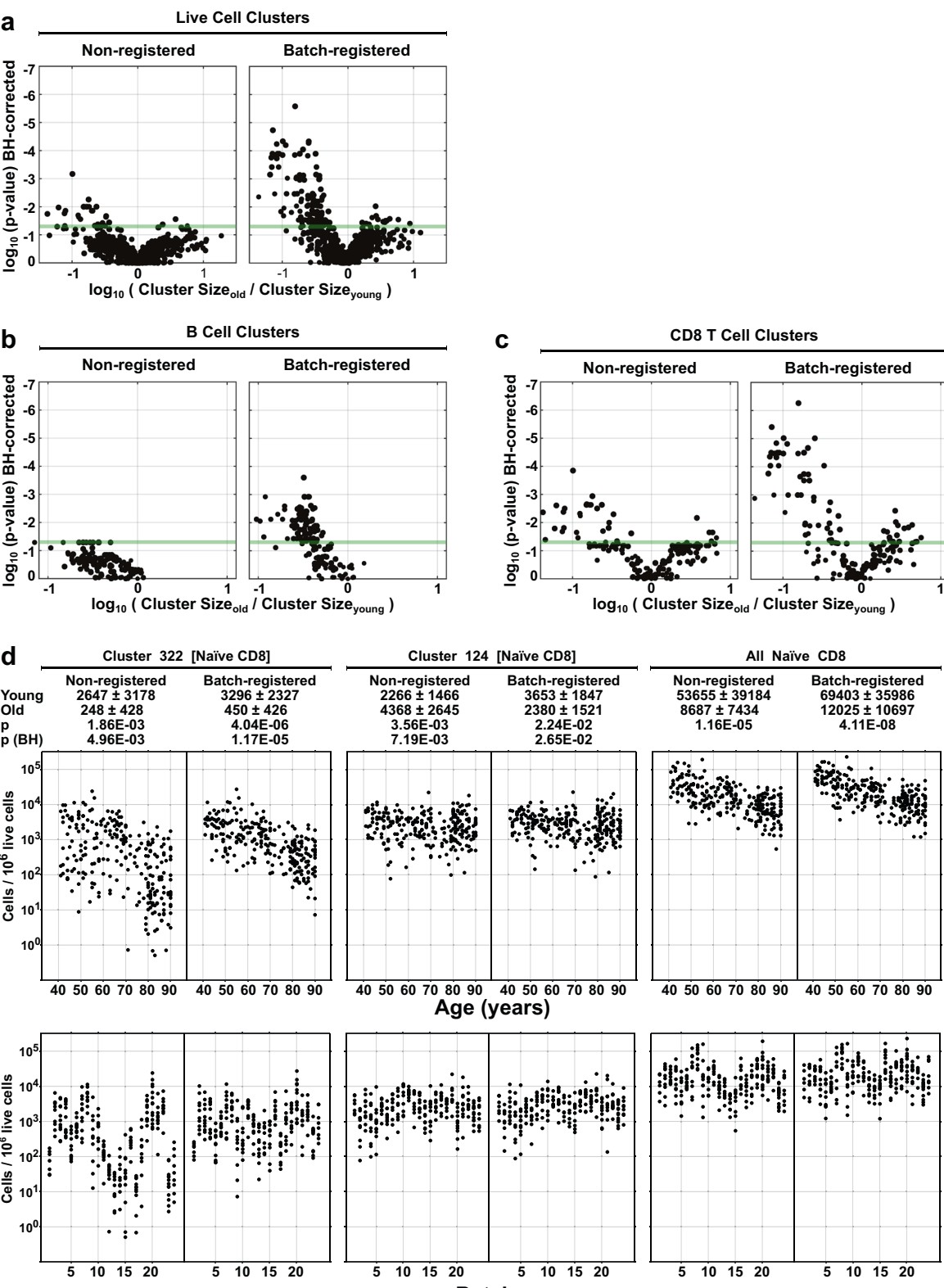

**Fig. 8 Evaluation of SWIFT clusters that change with age. a–c** The registered and non-registered samples described in Fig. 7 were analyzed by comparing the numbers of cells/cluster between the oldest 20 and youngest 20 subjects ($n = 20$ biologically independent samples per group). Each dot represents one cluster. $p$ Values were calculated by two-sided Wilcoxon test, followed by Benjamini–Hochberg correction for multiple measures and plotted against the ratio of the geometric means of the old and young groups. The colored line indicates $p = 0.05$. Clusters were classified by cluster gating (Supplementary Fig. 6a) into **a** live cell clusters, **b** B cell clusters, and **c** CD8 T cell clusters. In **d**, each dot represents one subject. Cluster sizes are sorted by age or batch for two selected naïve CD8 T cell clusters (124 and 322), and the sum of the cells in all 32 naïve CD8 T cell clusters for each subject. $p$ Values were derived from a two-sided Wilcoxon test of two subject groups (oldest vs. youngest) with $n = 20$ biologically independent samples per group.

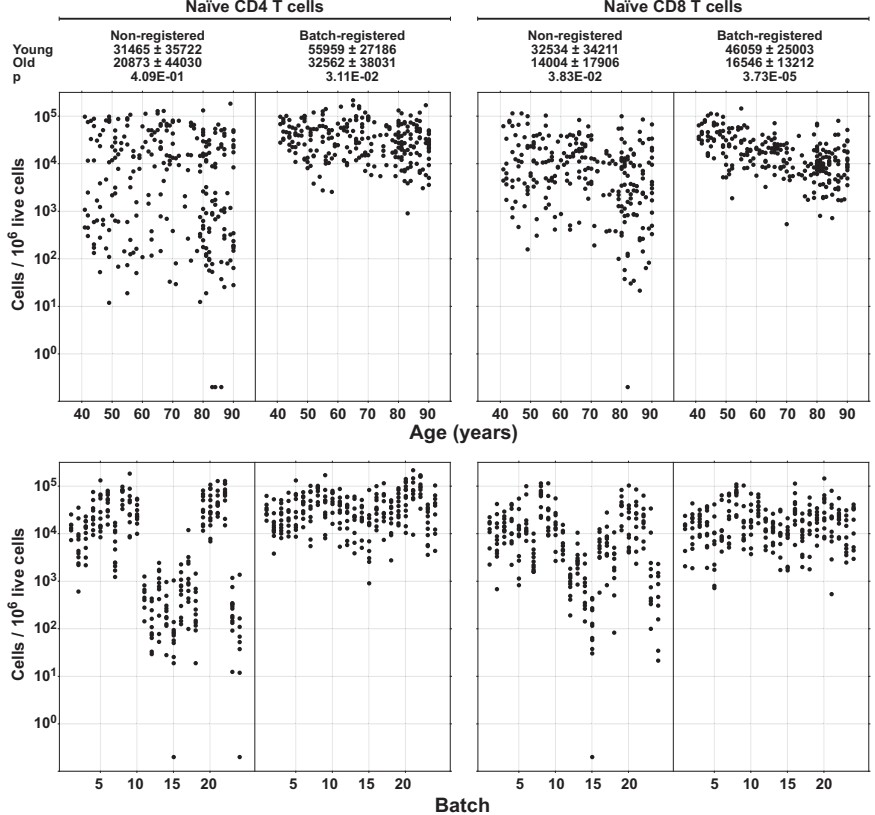

**Fig. 9 Manual analysis of the registered data.** The registered and non-registered samples described in Fig. 7 were analyzed manually using a single set of gates without any algorithmic assistance. The same gates (Supplementary Fig. 6b) were used to analyze both registered and non-registered data. Each dot represents one subject. Subpopulation sizes are sorted by age or batch for naive CD4 T cells or naive CD8 T cells. p Values were derived from a two-sided Wilcoxon test of two subject groups (oldest vs. youngest) with $n = 20$ biologically independent samples per group.

matches that cluster's centroid. The overall cluster movement vector is the combination of all the single-channel movement values. As with ICR, NDCR uses compound vectors for cells with multiple cluster memberships, registration is performed iteratively, and registered data is encoded and output as standard FCS files.

**Calculation of cluster weights for NDCR.** The cluster centroids used in generating the best-fit correction line are weighted according to size. During NDCR registration, corresponding clusters in the test sample and reference template may have different numbers of cells ($\kappa$). Owing to estimation uncertainty, $\kappa$ is the minimum size of the test sample or reference template cluster. Furthermore, we assume that centroid estimate accuracy improves with cluster size (i.e., law of large numbers applies) and that the centroid estimation errors relative to cluster covariances are consistent across clusters and experiments. We therefore fix our test statistic parameters to a standard deviation ($\sigma$) of 1.0, and a mean ($\mu$) of 1/15 but allow the $\kappa$ to vary. Our goal is to assign the cluster weight ($\omega$) based on the probability that a sample cluster is correctly assigned (within a tolerance of $\mu$) to its cluster in the reference. Setting the tolerance to a constant relative to $\sigma$ allows the probability to adjust according to the size of the smallest cluster between the reference or sample. Using this formulation (Eq. 2), we get the equation,

$$\omega = 1 - 2 \times t_{cdf}\left(-\frac{\mu}{\sigma\sqrt{\kappa}}, \max(1, \kappa - 1)\right) \qquad (2)$$

where $t_{cdf}$ is the Student's $t$ cumulative distribution function in MATLAB, with test statistic $-\frac{\mu}{\sigma\sqrt{\kappa}}$, and degrees of freedom $\max(1, \kappa - 1)$. The function is parameterized to produce a range of weights between ~0.1 and 1, such that large clusters (e.g., >10,000 cells) have equal weights of 1. Progressively smaller clusters have smaller weights (Supplementary Fig. 7). If there is no corresponding cluster in the test sample, then that cluster size is 0, and the calculated weight becomes 0 for that cluster. This would mean the zero-weighted cluster has no impact on the movement of any cells during registration. This setting is adjustable in the configuration file, and we recommend that it not be set <1/20 or >1/10. The default of 1/15 should be suitable for samples containing <20 million cells.

**Calculation of binning for NDCR.** To obtain stable bin movement vector estimates for NCDR, each bin should contain more than one cluster (ideally the more the better). We use the following heuristic (Eq. 3) to calculate the number of bins,

$$\text{Bins} = \left\lfloor 0.75 \times \sqrt{N} \right\rfloor \qquad (3)$$

where $N$ is the number of clusters. We further constrain the minimum and maximum number of bins to 2 and 20, respectively (Supplementary Fig. 8).

**Batch cluster registration.** To remove variation due to experimental batches, while maintaining as much biological variation as possible, swiftReg was extended to register batches. The overall strategy is to register the batches to each other as single units and apply the resulting batch-specific shifts to all individual samples in that batch.

A SWIFT cluster template is produced from an internal standard (or consensus sample) from a reference batch, and internal standards (or consensus samples) from each of the other batches are registered to the reference template. The resulting batch movement vectors are captured in a batch registration template. All individual samples in each batch are then registered using these batch-derived cluster movement vectors.

Selection of both reference and test samples is critical for successful batch registration. Two methods are appropriate: (1) If internal standards are available (e.g., if aliquots of a single sample were cryopreserved, and one aliquot was thawed to accompany each batch), then these should be used from both reference and test batches to derive the batch registration template. (2) If internal standards were not included, then a concatenate of all test samples in each batch should be used, provided each batch contained a similar distribution of samples from all the experimental groups.

**Statistics and reproducibility.** The SDY420 dataset consisted of 260 CyTOF flow cytometry samples from subjects with age varying between 41 and 90 years and contained no replicates (i.e., each sample was from a single subject). Two groups were identified from the 20 oldest and 20 youngest subjects ($n = 20$ biologically independent samples per group). p Values for all group-wise comparisons in Figs. 8 and 9 were calculated via a two-sided Wilcoxon test. p Values in the volcano plots in Figs. 8a–c were further corrected for multiple measures by the

Benjamini–Hochberg method. The three different datasets described in this paper show that our method reproducibly improves batch data consistency. We provide sufficient instruction to reproduce the same analyses.

**Reporting summary**. Further information on research design is available in the Nature Research Reporting Summary linked to this article.

## Data availability

The source datasets analyzed during the current study are available in the following repositories/accession codes: Fluorescence samples: JMW090 (8 samples) https://flowrepository.org/id/FR-FCM-ZZ8W; JMW090 (40 samples) https://flowrepository.org/id/FR-FCM-Z284; JMW092 https://flowrepository.org/id/FR-FCM-Z283; and HVTN080 https://flowrepository.org/id/FR-FCM-ZZ7U; CyTOF samples: SDY420 https://www.immport.org/shared/study/SDY420 and SDY420 Metadata (https://doi.org/10.5281/zenodo.3733491). All relevant data generated and analyzed are available from the authors upon reasonable request. Processed data underlying Figs. 2b, c; 3b, c; 5b; 8a–d; and 9 are available via the following links: Fig. 2 data (https://doi.org/10.5281/zenodo.3727063); Fig. 3 data (https://doi.org/10.5281/zenodo.3727074); Fig. 5 data (https://doi.org/10.5281/zenodo.3727077); Fig. 8 data (https://doi.org/10.5281/zenodo.3727082); and Fig. 9 data (https://doi.org/10.5281/zenodo.3727088).

## Code availability

swiftReg is included in the SWIFT package (https://doi.org/10.5281/zenodo.3704000), which is freely available for download at http://www.ece.rochester.edu/projects/siplab/Software/SWIFT.html

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

## Acknowledgements

We are grateful to Holden Maecker and Garry Fathman for sharing the batch information for the SDY420 aging dataset. This work was supported by the National Institutes of Health through UH2 AI 132339.

## Author contributions

J.A.R. and T.R.M. contributed to the conception and design of the work, creation of the new software, and analysis of data. S.A.Q. contributed to the acquisition and interpretation of data; G.S. contributed to the conception and design of the work and creation of the new software.

## Competing interests

The authors declare no competing interests.
