## [Peer Review File · Communications Biology]

Reviewers' comments:

Reviewer #1 (Remarks to the Author):

The study of Rebhahn et al. describes an elegant way to reduce batch variability to enhance downstream analysis. High dimensional cytometry data can provide deep insight into cellular phenotypes and functions, but technical variations from manual gating and experiments can confound the data analysis. The algorithm is a registration program that attempts to improve already available programs `fdaNorm` and `gaussNorm`. The authors make an interesting claim that `swiftReg` is able to reduce variation from batch effects, while also being able to study changes in biological variation. Such a method could be useful for validating observations from historical datasets.

They proved that `swiftReg` can reduce false positives of perforin expression in HVTN samples as described in previous report (ref 17). They also presented the selective correction of undesired variability by batch registration of SDY420 data. They briefly described limitations of `swiftReg` such as choice of references, but do not have adequate discussion on possible pitfalls of batch registration process. For example, SDY420 paper (ref 21) showed the considerable heterogeneity at every age in CD27+CD8+ T cell subsets and readers would wonder how the batch registration can carry out selective correction of undesirable variations without prior information on certain subsets. What if heterogeneous subsets like CD27+CD8+ T cell are monitored in a longitudinal study? What would be the proper template for the longitudinal study where monitoring heterogeneous subsets is critical? In Figure 6 they tried to address the selective correction of undesirable variations with synthetic data, but we suggest that they should add further discussion on the selective correction of variations with more data. Finally, some sentences and words are not completely clear, confusing the understanding of main advantages of `swiftReg`.

Specific comments

Introduction p3: "Several types of variability contribute to the changes in fluorescence intensity that are inherent in flow cytometry." They used CyTOF data, a panel of 27 markers, from SDY420 for the manuscript and need to rephrase the statement.

Introduction p5: "the method is robust to large changes in specific sub-populations, e.g. loss of CD4+ populations in AIDS.." There are no specific results supporting the statement except semi-synthetic example in Fig 3.

Results p6: "The heatmaps show the correlations of cluster centroids between samples.." Please describe the process to calculate correlations of cluster centroids.

Results p6: "The fine-grained analysis shows that certain parameters, e.g. CD45RA and CD4, contribute more strongly.." It's hard to agree with the statement based on Fig 1b. Plots are too small.

Results p7: "Therefore, we used the detailed information in the SWIFT cluster template.." Please describe what the detailed information is used to register samples at the sub-population level.

Results p10: "A cluster template is produced from an internal standard sample in the reference batch, and the internal standard from every other batch is registered to this template." Please briefly describe an internal standard sample in the reference batch. Reference 21, the SDY420 study does not describe the internal standard either.

Results p10: "the SDY420 study describes the CyTOF flow cytometry...". It looks like they only use CyTOF in the manuscript.

Results p10: "These samples were analyzed in 24 batches, with an experimental design that included an internal standard in each batch." How did the SDY420 study utilize internal standard to reduce batch effects in the original report?

Results p10-11 : "Note that heterogeneity WITHIN a batch was preserved, e.g. sample-specific variability in CD20 (first sample) or CD45RA (fourth sample)." This is not clear. Which figure is supporting the statement?

Results p11 : " However, many more clusters showed young/elderly differences in the registered data." Please describe more about those clusters. "Many more clusters" does not seem to be specific.

Methods p15: "Sample clusters are first ranked according to the cluster centroids.." Not clear of ranking the cluster centroids. Is it by the size of cluster?

Methods p15: "Because estimation uncertainty of centroids is inversely related to cluster size..." This might be right if sample clusters are close to template clusters. However, as authors mentioned in Introduction, a sample with loss of CD4+ population might still generate clusters as big as a sample with CD4+ population even though two clusters are completely different.

Figure 1a : Please describe the SWIFT clusters in details. How many clusters were used to calculate the R2? Are they from median fluorescence intensities of channels in clusters?

Figure 1b : Plots are too small and not informative. It is not clear what the purpose of Fig 1b is. It would be better to focus a couple of channels with more description. It's hard to correlate Fig 1a and Fig1b.

"...a ratio to the averages of the corresponding cluster in the samples from one subject and plotted on a log scale for all individual channels..." is not clear. What are "the averages of the corresponding cluster"?

Figure 2b : Please describe more about the number of clusters in reference templates. How many clusters are used to assign a sample to a template? Are there any concerns of artifacts by overfitting? As the dots are read as centroids, those are multi-dimensional. How were those dots projected to 1-dimensional axis?

Figure 3: Figure 3b shows that ICR works for both cases (cluster shifts and channel shifts) on semi-synthetic data, but real examples will make their point stronger. Please describe how to generate semi-synthetic data. Why does Cluster registration work better for Cluster Three than for Cluster One? Please describe more about Clusters.

Figure 6(a-f): Labels are missing for axes. Please clarify color codes. Also need to describe how to generate synthetic data. Are they from SDY420 dataset?

Figure 6(g-h) : Same issues as in Figure 1b.

Figure 6i : It would be extremely helpful if biological differences were described specifically. For example highlight and describe some clusters showing marked changes after batch registration.

Reviewer #2 (Remarks to the Author):

The authors describe a method for registration of cytometry data using the Swift clustering software. While the method is promising, I had a hard time following several of the results, figures and the actual methods used. Some clarification is needed before I can make a fully informed recommendation.

1. Could the Swift software with detailed instructions (or ideally, a reproducible analysis script) for replicating the results be provided for reviewers? Currently, it appears that Swift requires registration to use, which is not an ideal situation for anonymous review.
2. P6. For Figure 1A, the legend describes "Experiment" but the key shows "Days" - is this the same thing? Also, are the eight visits the small squares within the subject bar?
3. P6. In Fig 1B, there are low correlations for subjects 5 and 6 with subjects 1-4 in the CD45RA heatmap, but the corresponding variation in Figure 1B appears to be much less for CD45RA. Can you explain why?
4. P7. "Registration of fluorescence intensities between samples could be performed on the bulk data" - what does bulk data mean?
5. P7. Figure 2B and 2C, what is the interpretation of the axes? What does it mean that most

centroids (with two exceptions) are above the diagonal before registration?

6. P7. "In independent cluster registration (ICR), each cell is moved (in several iterations) according to the cluster movement vector of its cluster". What does "several iterations" mean? Who chooses the number of iterations, or is there some natural tolerance criterion for stopping? In the figure, the centroids appear to be simply aligned to the diagonal by horizontal shifting - how does iteration change the final outcome?

7. P7. ICR and NDCR are alternative ways of registering clusters, and it is not clear if there is a principled way to combine the suggested shifts. The manuscript gives some ad hoc suggestions, but it is not clearly explained under what circumstances we would prefer to give more weight to one over the other. On P8, it is suggested that "sequential registration by NDCR then ICR" gives the best results, but it is not clear that this generalizes beyond the semi-synthetic data set.

8. Given that Swift uses a Gaussian mixture model, presumably some cell subsets (especially those with a skewed distributions) are modeled by multiple clusters. Are there situations where the clusters belonging to the same cell subset might be shifted in discordant directions?

9. P8. In the lower panel of Figure 3A, the background contours get progressively smaller. Why is this if all cells are being shifted?

10. In Figure 3B, it is not clear why the shift size appears to be getting smaller from left to right, but the assignment accuracy improves from right to left.

11. For Figure 4, correlation is certainly improving, but it is not clear that this is not at the expense of eliminating true biological differences since there are 6 different subjects. A well-known paper suggests that cell subset "quality" or MFI gives important information - would registration hide such signals?

12. P9. Are batches 880 and 1053 pooled samples from multiple subjects in a batch?

13. P9. "As the major variation in this HVTN dataset was due to batch variation, we used NDCR to register all samples". What happens with ICR or the previous strategy of NNDCR followed by ICR?

14. P9. The cells of interest are described as CD4+ Perforin+, but the plots show granzyme B and perforin. Why not CD4 and perforin?

15. Figure 6i shows what seems to be a very large number of clusters. Do these correspond to groups of events that are meaningful cell subset clusters? If multiple clusters constitute the same subset (e.g. because the cell subset distribution is quite skewed or simply over-fitting), can spurious changes in ratio be induced by differential shifts?

16. How sensitive is the registration procedure to the reference sample selected? In a practical setting, how could one be sure that the reference "contain all the sub-populations present in any sample to be registered"?

17. A thought experiment suggests possible issues with the proposed method. Suppose we had a cell subset with a non-Gaussian distribution (say a uniform distribution), and fit a GMM twice to this (take one to be reference and one test). Then an arbitrary number of Gaussians may be used to approximate the distribution, but the individual components for each the test and reference mixture may be very different - even if the overall distribution is well approximated by the mixture. Registration of individual components Gaussians in the test to the reference could lead to severe distortion of the original cell subset distribution.

18. P15. Cells at the boundary of a cluster are most likely to have significant probabilities of belonging to two clusters. If the movement vectors of the two clusters are in divergent directions, won't these boundary cells move away from both parent clusters?

19. P15. The iterative procedure seems rather arbitrary. Why 25%, 33%, 50%, and 100%?

20. P15. I don't understand what how NDCR is done from the description. Do you bin small clusters with small cluster and large clusters with large clusters? Why use the minimum weight of each cluster between reference and test samples? Why 20 bins?

21. Supplemental Data:

1. "corresponding clusters in the test sample and reference template may have different numbers of cells". What if there are no corresponding clusters?

2. What is the statistical justification for the particular equation? Where does the number 15 in the denominator come from?

22. Could the authors generate histograms of each marker for the examples before/after

registration to provide an intuitive visualization of the effects of registration?

COMMSBIO-19-1059-T
Response To Reviewers

Reviewers' comments:

In response to many valid points made by both referees, we have extensively modified the manuscript, and in particular have tested several aspects of the registration process. This has resulted in a clearer understanding of the process, and we have made improvements in the method in line with this new data. Several new figures or figure parts have been added. The in-depth questions raised by both reviewers have led to substantial improvements, we have acknowledged this help in the Acknowledgements.

Reviewer #1 (Remarks to the Author):

The study of Rebhahn et al. describes an elegant way to reduce batch variability to enhance downstream analysis. High dimensional cytometry data can provide deep insight into cellular phenotypes and functions, but technical variations from manual gating and experiments can confound the data analysis. The algorithm is a registration program that attempts to improve already available programs `fdaNorm` and `gaussNorm`. The authors make an interesting claim that `swiftReg` is able to reduce variation from batch effects, while also being able to study changes in biological variation. Such a method could be useful for validating observations from historical datasets.

They proved that `swiftReg` can reduce false positives of perforin expression in HVTN samples as described in previous report (ref 17). They also presented the selective correction of undesired variability by batch registration of SDY420 data. They briefly described limitations of `swiftReg` such as choice of references, but do not have adequate discussion on possible pitfalls of batch registration process. For example, SDY420 paper (ref 21) showed the considerable heterogeneity at every age in CD27+CD8+ T cell subsets and readers would wonder how the batch registration can carry out selective correction of undesirable variations without prior information on certain subsets.

The internal standards in the SDY420 experiments comprised aliquots of a single sample from one subject, assayed in each batch. By calculating all the registration vectors required to register each cluster in the internal standards, registration can be performed without any pre-definition of populations of interest in the actual samples. In fact, it is critically important that the populations of interest are not pre-defined, so that the registered dataset can be explored in a non-biased manner. In experiments without internal standards, registration vectors can be calculated using consensus samples of each batch, provided that the representation of biological groups is approximately equal between batches. This information has now been reinforced in the Conclusions, first and fourth paragraphs.

What if heterogeneous subsets like CD27+CD8+ T cell are monitored in a longitudinal study? What would be the proper template for the longitudinal study where monitoring heterogeneous subsets is critical? In Figure 6 they tried to address the selective correction of undesirable variations with synthetic data, but we suggest that they should add further discussion on the selective correction of variations with more data.

We have expanded the description of the choice of proper registration template in the Conclusions (fourth paragraph), and added further discussion of the selective correction of batch variations, identifying multiple cell types, particularly with reference to the naïve CD8+ T cell subset mentioned in Table 2 of the original SDY420 paper as the most significant change associated with aging (the CD27+CD8+ subset correlated with CMV status). In general, major cell types defined by multiple clearcut markers (e.g. T cells are CD3+CD4+CD20-CD14-CD19-) can be identified accurately by the basic SWIFT clustering assignment step, so that substantial batch

variations can be handled reasonably by the basic SWIFT alone. Registration is more important for more subtle population differences, e.g. manually-gated naïve CD4 and CD8 T cells, or some of the high-resolution SWIFT clusters. This information has been added in the new Figures 8 and 9, described in three new paragraphs at the end of Results, and discussed in the Conclusions, particularly in the new second paragraph.

Finally, some sentences and words are not completely clear, confusing the understanding of main advantages of swiftReg.

Many clarifications have been added, as described in the detailed comments.

Specific comments

Introduction p3: “Several types of variability contribute to the changes in fluorescence intensity that are inherent in flow cytometry.” They used CyTOF data, a panel of 27 markers, from SDY420 for the manuscript and need to rephrase the statement.

This has been corrected.

Introduction p5: “the method is robust to large changes in specific sub-populations, e.g. loss of CD4+ populations in AIDS..” There are no specific results supporting the statement except semi-synthetic example in Fig 3.

This has been phrased more cautiously.

Results p6: “The heatmaps show the correlations of cluster centroids between samples..” Please describe the process to calculate correlations of cluster centroids.

A more detailed explanation has been added to Methods.

Results p6: “The fine-grained analysis shows that certain parameters, e.g. CD45RA and CD4, contribute more strongly..” It’s hard to agree with the statement based on Fig 1b. Plots are too small.

We have expanded CD45RA and CD4 plots as a separate figure element and describe the utility of the QC plot in more detail. Also, the y-axis units are now labeled: “log₁₀ (MFI sample / MFI reference)”

Results p7: “Therefore, we used the detailed information in the SWIFT cluster template..” Please describe what the detailed information is used to register samples at the sub-population level.

The detailed information refers to the model defined in the SWIFT cluster template. The model definition consists of cluster centroids (compensated, transformed means in each dimension), covariances (cluster shapes), and proportions (cluster sizes). This has been clarified in the Results.

Results p10: “A cluster template is produced from an internal standard sample in the reference batch, and the internal standard from every other batch is registered to this template.” Please briefly describe an internal standard sample in the reference batch. Reference 21, the SDY420 study does not describe the internal standard either.

Internal standards are aliquots of a single sample that were cryopreserved, and one aliquot was thawed to accompany each batch. A table of metadata has been added to the Supplementary data that indicates batches and internal standards.

Results p10: “the SDY420 study describes the CyTOF flow cytometry...”. It looks like they only use CyTOF in the manuscript.

We used fluorescence cytometry data for Figures 1, 2, 3a, 3b, 4 and 5. This information has been added to the Results and a figure legend.

Results p10: “These samples were analyzed in 24 batches, with an experimental design that included an internal standard in each batch.” How did the SDY420 study utilize internal standard to reduce batch effects in the original report?

The flow cytometry in the SDY420 study (Whiting et al) was not described in detail. FlowJo was used for manual "cell subset-specific gating" but the gating strategy was not specified, and even the antibody panel was described only as "[insert Ab list here]". The internal standards, or their use, were not described.

Results p10-11 : “Note that heterogeneity WITHIN a batch was preserved, e.g. sample-specific variability in CD20 (first sample) or CD45RA (fourth sample).” This is not clear. Which figure is supporting the statement?

Figure 7b now shows this information – the figure has been enlarged for clarity, and the relevant samples and channels indicated by arrows.

Results p11 : “ However, many more clusters showed young/elderly differences in the registered data.” Please describe more about those clusters.

As described above, Figure 6 has been expanded into Figure 6, 7, 8, and 9 to provide more information on specific cell types that showed enhancement of the significant differences between young and elderly. Cluster differences for B cell or CD8 T cell sub-populations are shown in Figure 8b and c, and naïve CD8 T cells are described in Figure 8d. This new data has resulted in a clearer explanation of the utility of swiftReg: batch variation in major populations defined by multiple clearcut markers is handled well by the main SWIFT algorithm, whereas swiftReg is more important for populations defined by subtle marker distinctions. This information has been added to the Results and Conclusions.

“Many more clusters” does not seem to be specific.

More detailed results at the cluster and traditional phenotype levels have been added to Figures 7 and 8, for the naïve CD8 T cell sub-populations that show major trends with age, as identified in the Whiting paper. We have also added manual gating analysis of the registered data in Figure 9, to show that the registered data also shows reduced batch effects by this analysis method.

Methods p15: “Sample clusters are first ranked according to the cluster centroids.” Not clear of ranking the cluster centroids. Is it by the size of cluster?

The rank refers to the cluster mean fluorescence intensity values within a channel of the sample being registered. This has been rephrased in the Methods to clarify.

Methods p15: “Because estimation uncertainty of centroids is inversely related to cluster size...” This might be right if sample clusters are close to template clusters. However, as authors mentioned in Introduction, a sample with loss of CD4+ population might still generate clusters as big as a sample with CD4+ population even though two clusters are completely different.

The reference, and the sample to be registered, are assigned to the same cluster template. If sizes of a cluster are disparate between reference and sample, the smaller size is used to calculate the weight for NDCR. This is explained in the expanded section in Supplemental Data.

Figure 1a : Please describe the SWIFT clusters in details. How many clusters were used to calculate the R²?

All 104 clusters were used to calculate Pearson correlation coefficients. This has been added to the figure legend.

Are they from median fluorescence intensities of channels in clusters?

Values in the heatmaps and QC plots are the medians of compensated fluorescence intensity values for each cluster that are then ArcSinh-transformed. This has been added to the Methods.

Figure 1b : Plots are too small and not informative. It is not clear what the purpose of Fig 1b is. It would be better to focus a couple of channels with more description. It's hard to correlate Fig 1a and Fig1b.

We have expanded on a subset of samples for two channels (new Figure 1b), as suggested. We also emphasize that QC plots are intended to facilitate a fast qualitative assessment of variability across many samples.

“...a ratio to the averages of the corresponding cluster in the samples from one subject and plotted on a log scale for all individual channels...” is not clear. What are “the averages of the corresponding cluster”?

Each dot within a channel-row represents one cluster from one sample. A cluster's vertical-position within the channel-row is the log ratio of its MFI to the reference standard MFI in that channel. The standard MFI's for each cluster were the averages of the MFI's for that cluster in all eight samples from subject 5. This has been added to the Results and figure legend (see also reviewer 2, point 3).

Figure 2b : Please describe more about the number of clusters in reference templates.

The reference template contains 184 clusters. This has been added to the figure legend.

How many clusters are used to assign a sample to a template?

When a sample is assigned to a template, each cell in the sample is assigned to one of the cluster descriptions in the model (cluster template). So the number of clusters depends on the template. For cells that have a significant probability of belonging to two or more clusters, fractional assignment is used. This information is described in the original SWIFT references, and has now been added in more detail to the Methods.

Are there any concerns of artifacts by overfitting?

The initial template is important: To avoid overfitting to a particular sample, consensus concatenates for e.g. batches are normally used. If the experimental design has one internal standard, then that sample should be used to prevent overfitting that could be caused by any of the samples in a particular batch. Overfitting can also be considered for the significance of the high number of clusters identified by SWIFT. Our criterion for separating sub-populations (no multimodality in any combination of dimensions) is intuitively reasonable, but results in an increased number of sub-populations when compared to traditional definitions of populations in sequential two-dimensional gates. To partly address this issue and to simplify the cluster complexity (new Figures 8, 9), we have aggregated the high resolution SWIFT clusters into traditional cell types (e.g. naïve CD4 T cells, naïve CD8 T cells) and shown the effect of registration both with SWIFT clustering, as well as manual gating. This has been described in the new figures, and in the text in the Results.

As the dots are read as centroids, those are multi-dimensional. How were those dots projected to 1-dimensional axis?

Although centroids can refer to uni-dimensional data, we have now referred to these particular values as mean fluorescence intensities in Figure 2(b,c) to clarify that they refer to one dimension.

Figure 3: Figure 3b shows that ICR works for both cases (cluster shifts and channel shifts) on semi-synthetic data, but real examples will make their point stronger. Please describe how to generate semi-synthetic data.

We have expanded the description of the creation of semi-synthetic samples in their own section in the Methods. We have also added a new analysis on real data (the new Figure 3c) that evaluates the performance of NDCR, ICR, and NDCR/ICR, analyzing the completeness of the position updates (partial vs full) as the number of iterations is varied.

Why does Cluster registration work better for Cluster Three than for Cluster One? Please describe more about Clusters.

Each cluster is potentially interesting, but is difficult to establish ground truth for all the multi-dimensional positions of each cluster. So the most important information, we believe, is that the biological information is seen with greater clarity after registration, e.g. figures 5, 7, 8, 9. To further address this point, we have broadened the analysis by using real data and calculating the RMSE of the sample:reference positions to incorporate information about the completeness of registration over all clusters. This allowed a quantitative assessment of the effect of different numbers of iterations, as well as the effect of partial or full registration at each iteration. These results are presented in the new Figure 3c (including only real data), and have led to a modification in our recommendation for optimal registration. As described in more detail in point 7, reviewer 2, we now recommend a total of four iterations of registration using full (not partial) movement vectors. This information has been added to the Results and Conclusions.

Figure 6(a-f): Labels are missing for axes. Please clarify color codes. Also need to describe how to generate synthetic data. Are they from SDY420 dataset?

Axis labels have been added and the generation of the synthetic data has been explained in the figure legend. The variations represented by the colors in each panel are now described in the legend.

Figure 6(g-h) : Same issues as in Figure 1b.

We have expanded the figure to give extra clarity, and added y-axis labels (now Figure 7).

Figure 6i : It would be extremely helpful if biological differences were described specifically. For example highlight and describe some clusters showing marked changes after batch registration.

As described above, the new Figures 8 and 9 have been added, and the new results described in the last three paragraphs of the Results, exploring specific populations in more detail.

Reviewer #2 (Remarks to the Author):

The authors describe a method for registration of cytometry data using the Swift clustering software. While the method is promising, I had a hard time following several of the results, figures and the actual methods used. Some clarification is needed before I can make a fully informed recommendation.

1. Could the Swift software with detailed instructions (or ideally, a reproducible analysis script) for replicating the results be provided for reviewers? Currently, it appears that Swift requires registration to use, which is not an ideal situation for anonymous review.

We have deposited a working model (in an anonymous file sharing service) which takes specified data, performs registration, and shows a simple graphical output of the before/after results. The associated Demo_README.pdf file describes the platform, operating system and software versions, and related instructions for use. MATLAB is required.

Below are two links for WeTransfer, which will remain active for 7 days or 3 downloads, whichever comes first. The package is about 300 MB, and to unzip please use the password 'demotest'. WeTransfer has a policy not to sell the data they track, and it is anonymized when used internally for legitimate business purposes (consistent with CodeOcean or GitHub too). Their terms of service state:

"To share your Content you need to upload it and provide us with (a limited number of) email addresses of recipient(s) ("email transfer") or choose to distribute a download link yourself ("link transfer"). If you use link transfer you will not be informed of any downloads by others."

We set up two identical links, one for each reviewer. The URLs for the "link transfer" are:

Reviewer 1: <https://we.tl/t-jSrqiAyU06>

Reviewer 2: <https://we.tl/t-IkKC9GO7ei>

These links expire after 7 days – we can establish extra links if necessary.

2. P6. For Figure 1A, the legend describes "Experiment" but the key shows "Days" - is this the same thing?

Yes, this has been changed to Assay Days in all cases.

Also, are the eight visits the small squares within the subject bar?

Yes, this information has been added to the legend, and for consistency all occurrences of 'visits' have been changed to 'bleeds'.

3. P6. In Fig 1B, there are low correlations for subjects 5 and 6 with subjects 1-4 in the CD45RA heatmap, but the corresponding variation in Figure 1B appears to be much less for CD45RA. Can you explain why?

We have inserted an expanded view of CD45RA and CD4 in the new Figure 1b. We have also specified that the average of subject #5 samples was used as the reference standard for the comparisons in Figs 1b and 1c. Thus subjects 5 and 6 show low dispersion around the center line (i.e. are similar to the reference) as expected from Fig. 1a, and subjects 1-4 show more variation. The description of the center line, which represents the 1:1 log-ratio between sample MFI and the standard, has also been expanded in the Results and figure legend.

4. P7. "Registration of fluorescence intensities between samples could be performed on the bulk data" - what does bulk data mean?

Bulk data refers to the fluorescence intensity values of all cells within a channel. If registration is performed on data in one channel as a single histogram, shifts in intensity due to the possibility diagrammed in Figure 2a could cause errors. This has been rephrased in the Results for clarity.

5. P7. Figure 2B and 2C, what is the interpretation of the axes?

An explanation of the axis values (mean fluorescence values in the CD3 channel) has been added to the legend.

What does it mean that most centroids (with two exceptions) are above the diagonal before registration?

The staining in the test sample was slightly lower, in this channel, than the reference sample. This comment has been added to the Results.

6. P7. "In independent cluster registration (ICR), each cell is moved (in several iterations) according to the cluster movement vector of its cluster". What does "several iterations" mean? Who chooses the number of iterations, or is there some natural tolerance criterion for stopping?

In response to this issue, we have performed extensive testing of the effectiveness of different iterations of NDCR and ICR. This has led to a clearer understanding of the registration process, and resulted in minor changes in the default registration settings, and our recommendations for optimal use of ICR and NDCR. The results are now presented in Figure 3c, and described in more detail below (see point 7).

In the figure, the centroids appear to be simply aligned to the diagonal by horizontal shifting - how does iteration change the final outcome?

Figure 2 originally showed the concept of how the cluster centroids are moved - we have now changed Figure 2 to show real registration results, so there is not perfect alignment, consistent with Figures 3 and 7. As the cells are re-assigned after every iteration (see methods), the cell makeup of each cluster will vary slightly with iterations. This method was designed to allow initial mis-assignment on some cells, particularly on cluster fringes, to be corrected in subsequent cycles, thus improving the estimate of the medians.

7. P7. ICR and NDCR are alternative ways of registering clusters, and it is not clear if there is a principled way to combine the suggested shifts. The manuscript gives some ad hoc suggestions, but it is not clearly explained under what circumstances we would prefer to give more weight to one over the other. On P8, it is suggested that "sequential registration by NDCR then ICR" gives the best results, but it is not clear that this generalizes beyond the semi-synthetic data set.

As described above, we have performed further testing, and included Figure 3c to show the effects of NDCR vs ICR; partial versus full iterations; and the number of iterations. NDCR is designed to be most effective for channel shifts, e.g. PMT amplification values, because all clusters contribute to the assessment, and this method preserves variations between samples in individual clusters. In contrast, ICR is necessary for cell-type-specific changes, e.g. multiple divergent alterations in cell properties due to mistreatment of cells during a particular batch of the experiment. ICR is particularly useful for batch registration using internal standard samples because batch variation is designed to preserve individual sample variation (that could contain biological information). As the relative contributions of channel-specific and cluster-specific variation cannot be determined for real-world datasets, theoretically the combination of methods should be optimal. This is borne out by the results of the semi-synthetic datasets, particularly showing that the joint method is normally as good as, or better than the two single methods (Figure 3b). This is further confirmed using real data in the new Figure 3c, showing that the RMSE of all cluster sizes is reduced most effectively by the combination of NDCR and ICR. It is important to emphasize that swiftReg is a practical tool – it will not remove all artefactual variation, but we have found that it offers some degree of improvement for all the datasets we have examined. Although we have designed swiftReg to deal with known types of variation in flow data, we do not know the origin of all types of variation, so the practical test of data improvement (better resolution of biological groups, e.g. in Figures 7, 8 and 9) is important for assessing swiftReg utility. Figure 3c also shows the effects of varying the number of iterative registration cycles, and using partial movement at each step (our original method) versus complete movement at each iteration. Full movement at each step resulted in more rapid reduction of the RMSE, and so we now use full movement as the default behavior during registration. The number of iterations, and the choice of NDCR or ICR, are still user-configurable.

Our new recommendations (added to Conclusions) are:

- For batch registration with internal standards, use NDCR followed by ICR.*
- To remove all variation between samples, e.g. enumerating cytokine-producing cells without preserving minor positional information, use NDCR followed by ICR on individual samples.*
- To remove channel-specific effects (e.g. staining or PMT voltage changes) in the absence of internal controls, and to preserve minor positional information, use NDCR only.*

- *If some batches contain sub-populations that are completely absent in other batches (or their controls), use NDCR only.*
- *In each case, a total of four iterations are recommended, as a compromise between speed and completeness.*

8. Given that Swift uses a Gaussian mixture model, presumably some cell subsets (especially those with a skewed distributions) are modeled by multiple clusters. Are there situations where the clusters belonging to the same cell subset might be shifted in discordant directions?

SWIFT uses a sequential clustering/splitting/merging approach to attempt to deal with the issue of skewed distributions. As mentioned by the referee, it is possible that two cell sub-populations may be so close in one sample that they appear to be a single sub-population, whereas in a second sample (e.g. with better staining in one crucial parameter) the sub-population may be bimodal. There are two possibilities – if the reference is bimodal, there will be two clusters, and the brighter shoulder of the unimodal sample sub-population will be preferentially assigned to the brighter of the two clusters in the reference, resulting in stretching of the shoulder of the unimodal sample sub-population to brighter values. Alternatively, if the reference has the unimodal sub-population, swiftReg should assign both dim and bright cells in the sample sub-population to this cluster, resulting in reduction of the MFI of the brighter sub-population. Either possibility would improve the registration between reference and sample.

9. P8. In the lower panel of Figure 3A, the background contours get progressively smaller. Why is this if all cells are being shifted?

The data were ArcSinh-transformed (see axis), and thus the scaled values are compressed as the magnitude decreases. This information has been added to the figure legend.

10. In Figure 3B, it is not clear why the shift size appears to be getting smaller from left to right, but the assignment accuracy improves from right to left.

Yes, this was ambiguous, the shift increases from left to right. The increments are now expressed as magnitudes, i.e. numbers increase to indicate larger deviations between non-scaled and scaled values.

11. For Figure 4, correlation is certainly improving, but it is not clear that this is not at the expense of eliminating true biological differences since there are 6 different subjects. A well-known paper suggests that cell subset "quality" or MFI gives important information - would registration hide such signals?

Absolutely, and this has been addressed in the Conclusions. New text in the Results and Conclusions emphasizes that registration has to be used cautiously, particularly ICR that has the ability to remove most MFI variation if applied directly to individual samples from different subjects. In contrast, ICR is very useful in batch registration, in which the biological variation can be maintained and emphasized.

See also answer to point 7.

12. P9. Are batches 880 and 1053 pooled samples from multiple subjects in a batch?

Results are pooled within each batch and stimulation for display only – 40 subjects total in each batch. This has been added to the figure legend, and Figure 5a shows the breakdown of the display pooling.

13. P9. "As the major variation in this HVTN dataset was due to batch variation, we used NDCR to register all samples". What happens with ICR or the previous strategy of NNDCR followed by ICR?

Some batches in this dataset lacked positively stained, rare populations for stimulated samples; thus, certain populations existed in some batches that were not present in others. As a result, ICR was not used so as to avoid potential mis-assignment, and NDCR alone was used to register all samples. However, for batch registration, the problem is reduced with internal control samples, as these should always contain all sub-populations. We have strengthened the advice regarding the choice of method to emphasize the importance of missing populations (paragraph 5, Conclusions).

14. P9. The cells of interest are described as CD4+ Perforin+, but the plots show granzyme B and perforin. Why not CD4 and perforin?

The original paper showed Perf and GZB, so we wanted to show comparable data to their Figure 5a. Cells were already gated on CD4. We can show CD4/Perf, but would prefer to keep direct comparability to the reference.

15. Figure 6i shows what seems to be a very large number of clusters. Do these correspond to groups of events that are meaningful cell subset clusters? If multiple clusters constitute the same subset (e.g. because the cell subset distribution is quite skewed or simply over-fitting), can spurious changes in ratio be induced by differential shifts?

This figure shows CyTOF data, in which the large number of markers results in a large number of sub-populations defined by our multidimensional unimodality criterion (see reference 2). Because the shapes of sub-populations in real samples can vary in unknown ways, it is difficult or impossible to be sure that registration will improve the correspondence of ALL sub-populations without any distortions. We have emphasized in the Conclusions that swiftReg can, in aggregate, improve the overall detection of biological differences in datasets, but we expect that it will be possible to find occasional sub-populations that show anomalous results.

We have provided further details of the clusters that show differences with aging, in the new Figure 8. Also see reply to point 8.

16. How sensitive is the registration procedure to the reference sample selected? In a practical setting, how could one be sure that the reference "contain all the sub-populations present in any sample to be registered"?

This issue has now been expanded (see point 7 above), with more specific recommendations about the reference sample and the choice of method. NDCR+ICR results in better overall alignment, but is best used in batch registration when internal control samples are available, because the number of sub-populations should be the same in the controls in each batch. This issue has been addressed in the Conclusions, paragraphs 3, 4 and 5.

17. A thought experiment suggests possible issues with the proposed method. Suppose we had a cell subset with a non-Gaussian distribution (say a uniform distribution), and fit a GMM twice to this (take one to be reference and one test). Then an arbitrary number of Gaussians may be used to approximate the distribution, but the individual components for each the test and reference mixture may be very different - even if the overall distribution is well approximated by the mixture. Registration of individual components Gaussians in the test to the reference could lead to severe distortion of the original cell subset distribution.

The referee is correct that matching clusters between two separate clusterings could lead to problems, particularly in the context of registration. We do not do this. We use one template that contains a model derived from only the reference sample, then assign cells from the test sample to the reference template model. The test sample never has a model of its own, its cells are only assigned to clusters in the reference template. This has been clarified in the Methods and Results.

18. P15. Cells at the boundary of a cluster are most likely to have significant probabilities of belonging to two clusters. If the movement vectors of the two clusters are in divergent directions, won't these boundary cells move away from both parent clusters?

Yes, in ICR this will happen, i.e. the adjustment is elastic. We believe that this is an important positive feature of ICR, because it ensures a smooth transition between different regions of the data, i.e. adjacent cells will show similar shifts.

19. P15. The iterative procedure seems rather arbitrary. Why 25%, 33%, 50%, and 100%?

We have explored this in more detail and have expanded figure 3 with a new analysis. We compared the effects of partial versus full position-updates, as well as varying the number of iterations. As a result, we have changed our recommendation to only do full (i.e. 100%) position-updates per iteration, and we confirm that registration improves with multiple iterations. This has been added to the Results (Figure 3c), the Methods have been modified, and we have refined our recommendation in the Conclusions.

20. P15. I don't understand what how NDCR is done from the description. Do you bin small clusters with small cluster and large clusters with large clusters? Why use the minimum weight of each cluster between reference and test samples? Why 20 bins?

We apologize for the confusing description. To clarify, clusters within a channel are first ranked in ascending order according to the cluster mean fluorescence intensity values of the sample being registered. Then cluster weights are calculated. Then clusters within a channel are binned such that the cluster weight per bin is a percentage of the total cluster weights, thus for example, with 20 bins each bin would contain 5% by weight of clusters. Also, the number of bins used is a function of the number of clusters. The mention of minimum weight referred indirectly to the calculation of the cluster weights based on the minimum size between a test and reference cluster. The NDCR description has been clarified in the Methods, and the cluster weight and binning calculations have also been expanded in the Supplemental section.

21. Supplemental Data:

1. "corresponding clusters in the test sample and reference template may have different numbers of cells". What if there are no corresponding clusters?

As described above, a single reference cluster template is used. If there are no corresponding cells in a particular cluster in the test sample, then that cluster size is 0, and the calculated weight becomes 0. This also means that the cluster would have no impact on the movement of any cells during registration. This point has been added to the Supplemental data.

2. What is the statistical justification for the particular equation? Where does the number 15 in the denominator come from?

We assume that centroid estimate accuracy improves with cluster size (i.e. Law of Large Numbers applies), and that the centroid estimation errors relative to cluster covariances are consistent across clusters and experiments. We therefore fix our test statistic parameters: standard deviation (σ) of 1.0, and mean (μ) of 1/15, but allow the cluster size (κ) to vary. Our goal is to assign the cluster weight (ω) based on the probability that a sample cluster is correctly assigned (within a tolerance of μ) to its cluster in the reference. Setting the tolerance to a constant relative to σ allows the probability to adjust according to the size of the smallest cluster between the reference or sample. Using this formulation we get the equation,

$$\omega = 1 - 2 \times \text{tcdf}\left(-\frac{\mu}{\sigma\sqrt{\kappa}}, \max(1, \kappa - 1)\right)$$

We realize the choice of 1/15 is arbitrary, so we have exposed that variable in the configuration file, and recommend that it not be set below 1/20 or greater than 1/10. The default of 1/15 should be suitable for samples containing less than 20 million cells. We have included this additional information in the Supplemental Data.

22. Could the authors generate histograms of each marker for the examples before/after registration to provide an intuitive visualization of the effects of registration?

Histograms have now been included in the Supplemental Data, for five batches of SDY420 with the most variation, for each channel, before and after registration. These give a clear visual representation of the improved alignment of the channels after registration, although the bulk data does not show the full improvement at the cluster level, as explained in the Results (in the description of Figure 2, first paragraph).

REVIEWERS' COMMENTS:

Reviewer #1 (Remarks to the Author):

Reviewer's overall comments:

Authors addressed the questions and concerns raised by reviewers in the revised manuscript. They also provided clearer figures and detailed legends to present their ideas and results.

Specific comments:

Why does Cluster registration work better for Cluster Three than for Cluster One? Please describe more about Clusters.

Author response:

Each cluster is potentially interesting, but is difficult to establish ground truth for all the multidimensional positions of each cluster. So the most important information, we believe, is that the biological information is seen with greater clarity after registration, e.g. figures 5, 7, 8, 9. To further address this point, we have broadened the analysis by using real data and calculating the RMSE of the sample:reference positions to incorporate information about the completeness of registration over all clusters. This allowed a quantitative assessment of the effect of different numbers of iterations, as well as the effect of partial or full registration at each iteration. These results are presented in the new Figure 3c (including only real data), and have led to a modification in our recommendation for optimal registration. As described in more detail in point 7, reviewer 2, we now recommend a total of four iterations of registration using full (not partial) movement vectors. This information has been added to the Results and Conclusions.

Reviewer #1:

I thought there were three clusters in synthetic data (Fig 3a) assigned to three reference clusters. Each cluster shifted away from original coordinates and were moved by NDCR/ICR (Fig 3b). But in Methods, p18, "one cluster was selected for tracking" which probably indicates blue circle in Fig 3a. Can you label three clusters in Fig 3a if I understand the figures correctly? At 3.2 magnitude of deviation for channel scaled example (bottom plots in Fig3a) all three clusters were shifted from template clusters, but more cells were correctly assigned to template clusters than cluster scaled example. Can you provide some explanation?

Additional comments:

Methods p21; "Then a smooth spline curve is fitted across the average bin intensities (x-axis) and bin movement vector (y-axis), and attenuated to zero for outlier regions with no clusters" Previous paragraphs described the registration process with a channel and "bin movement scalar" seems more appropriate than "bin movement vector".

Are all cells assigned to reference clusters? If not, those unassigned cells might be direct results from technical variations or biological variations.

Methods p19: In Calculation of cluster movement vectors section, I am just wondering if there are any relationships between any parameters in sample cluster assignment (to the reference GMM) and swiftReg performance.

In Figure 8d, I expect "All naïve CD8" plots include the data from plots of cluster 322 and cluster 124, but that's not the case. Can you provide some details about those plots?

What are the difference between "All naïve CD8" in Figure 8d and "Naïve CD8 T cells" in Figure 9? Are the plots in Figure 8d from Cluster gating shown in Supp. Figure 8?

Supp. Figure 6 and Figure 7 The histograms are very informative. Most of peaks across channels are well aligned after registration. Can you take a look at CD28 channels for some subjects such as S137502? Clearly more CD28 positive cells are shown after registration.

Reviewer #2 (Remarks to the Author):

The authors have provided a detailed response to my questions and revised the manuscript fairly extensively. The procedure is now clearer and comes with more explicit guidelines, and at some caveats are now discussed. Many figures that were confusing have been improved.

The method proposed has innovative ideas, and appears to work reasonably well in practice. There are a few drawbacks to the method proposed - first, the algorithm is developed for a proprietary software (Matlab) making it less accessible, and second, the approach is heuristic in nature and may be brittle outside the range of samples tested.

On balance, I think it is a useful contribution to the cytometry community that does a reasonably good job of addressing a difficult challenge - that of calibration or alignment of multi-batch samples.

REVIEWERS' COMMENTS (Second round):

Reviewer #1 (Remarks to the Author):

Reviewer's overall comments:

Authors addressed the questions and concerns raised by reviewers in the revised manuscript. They also provided clearer figures and detailed legends to present their ideas and results.

Specific comments (previous review):

Why does Cluster registration work better for Cluster Three than for Cluster One? Please describe more about Clusters.

Author response (previous review):

Each cluster is potentially interesting, but is difficult to establish ground truth for all the multidimensional positions of each cluster. So the most important information, we believe, is that the biological information is seen with greater clarity after registration, e.g. figures 5, 7, 8, 9. To further address this point, we have broadened the analysis by using real data and calculating the RMSE of the sample:reference positions to incorporate information about the completeness of registration over all clusters. This allowed a quantitative assessment of the effect of different numbers of iterations, as well as the effect of partial or full registration at each iteration. These results are presented in the new Figure 3c (including only real data), and have led to a modification in our recommendation for optimal registration. As described in more detail in point 7, reviewer 2, we now recommend a total of four iterations of registration using full (not partial) movement vectors. This information has been added to the Results and Conclusions.

Reviewer #1:

I thought there were three clusters in synthetic data (Fig 3a) assigned to three reference clusters. Each cluster shifted away from original coordinates and were moved by NDCR/ICR (Fig 3b). But in Methods, p18, "one cluster was selected for tracking" which probably indicates blue circle in Fig 3a. Can you label three clusters in Fig 3a if I understand the figures correctly? At 3.2 magnitude of deviation for channel scaled example (bottom plots in Fig3a) all three clusters were shifted from template clusters, but more cells were correctly assigned to template clusters than cluster scaled example. Can you provide some explanation?

Yes, this was not clear, some of the information was omitted. The example in Fig. 3a shows only one cluster, which is cluster 1 in Figure 3b. The other two clusters in 3b were each from separate samples, so these can't be displayed on the same plots in 3a. We have modified the Results and figure legend to explain this in more detail.

Additional comments:

Methods p21; "Then a smooth spline curve is fitted across the average bin intensities (x-axis) and bin movement vector (y-axis), and attenuated to zero for outlier regions with no clusters" Previous paragraphs described the registration process with a channel and "bin movement scalar" seems more appropriate than "bin movement vector".

We have re-examined this point in detail. The error identified by the referee was one of several locations where we had not discriminated between vector and scalar values. As a result, we have made several small adjustments in the Methods, the Results, the legend to Figure 2, and Figure 2 itself. These changes explicitly designate single-channel and multichannel movements. We checked with biologist colleagues, and found that 'scalar' is not a familiar term, so we have used 'value' instead of 'scalar' for all one-dimensional shifts.

Are all cells assigned to reference clusters? If not, those unassigned cells might be direct results from technical variations or biological variations.

Yes, all cells are assigned to clusters. This was explained in the original SWIFT papers, and we have now added extra explanation in the Methods.

Methods p19: In Calculation of cluster movement vectors section, I am just wondering if there are any relationships between any parameters in sample cluster assignment (to the reference GMM) and swiftReg performance.

The assignment step does not have any tunable parameters. However, when constructing the reference GMM it is very important to consider several properties of the individual experiments. For example, if the reference GMM is constructed from a relatively low number of cells, a rare sub-population of cells may not be recognized as a separate cluster. This is part of the larger issue of ensuring that the reference GMM contains adequate representation of all cell sub-populations in the samples to be registered. These considerations are addressed in both the Results, and Discussion (paragraphs 4 and 5).

In Figure 8d, I expect “All naïve CD8” plots include the data from plots of cluster 322 and cluster 124, but that’s not the case. Can you provide some details about those plots?

In the “All Naïve CD8” panel in Figure 8d, each dot shows the sum of the number of cells in all 32 naïve CD8 T cell clusters for one subject. Thus the number of data points is the same as in the other panels. This information was in the figure legend, but we have now rephrased for clarity.

What are the difference between “All naïve CD8” in Figure 8d and “Naïve CD8 T cells” in Figure 9? Are the plots in Figure 8d from Cluster gating shown in Supp. Figure 8?

Figure 8 shows cluster data, and Figure 9 shows manual gating results. This information was already in the figure titles – we have now added SWIFT cluster to the title of Figure 8 to emphasize that these are clusters derived algorithmically.

Yes, the Cluster gating is shown in Supplementary Figure 8 (now referenced as 8a). We found a mistake in the related reference to the manual gating figure – this has now been corrected from Supp Fig 9 to Supp Fig 8b.

Supp. Figure 6 and Figure 7 The histograms are very informative. Most of peaks across channels are well aligned after registration. Can you take a look at CD28 channels for some subjects such as S137502? Clearly more CD28 positive cells are shown after registration.

Yes, this behavior is expected because swiftReg works at the level of individual clusters, not the bulk channel data. As shown in Figure 2, bulk data profiles can be affected both by changes in fluorescence intensity, as well as population size changes. As shown in the QC plots (e.g. Fig. 7) some clusters require larger registration shifts than others, so the changes noticed by the reviewer are to be expected.

Reviewer #2 (Remarks to the Author):

The authors have provided a detailed response to my questions and revised the manuscript fairly extensively. The procedure is now clearer and comes with more explicit guidelines, and at some caveats are now discussed. Many figures that were confusing have been improved.

The method proposed has innovative ideas, and appears to work reasonably well in practice. There are a few drawbacks to the method proposed - first, the algorithm is developed for a proprietary software (Matlab) making it less accessible, and second, the approach is heuristic in nature and may be brittle outside the range of samples tested.

On balance, I think it is a useful contribution to the cytometry community that does a reasonably good job of addressing a difficult challenge - that of calibration or alignment of multi-batch samples.